# Sound and Complete Causal Identification with Latent Variables Given Local Background Knowledge

**Tian-Zuo Wang, Tian Qin, Zhi-Hua Zhou**

National Key Laboratory for Novel Software Technology,
Nanjing University, Nanjing, 210023, China.
{wangtz, qint, zhouzh}@lamda.nju.edu.cn

## Abstract

Great efforts have been devoted to causal discovery from observational data, and it is well known that introducing some background knowledge attained from experiments or human expertise can be very helpful. However, it remains unknown that *what causal relations are identifiable given background knowledge in the presence of latent confounders*. In this paper, we solve the problem with sound and complete orientation rules when the background knowledge is given in a *local* form. Furthermore, based on the solution to the problem, this paper proposes a general active learning framework for causal discovery in the presence of latent confounders, with its effectiveness and efficiency validated by experiments.

## 1 Introduction

Causality has attracted tremendous attention in recent years, for its application on explainability [1], fairness [2, 3, 4], decision [5, 6, 7, 8, 9], and so on. In Pearl's causality framework [10], one important problem is *causal discovery*, *i.e.*, learning a causal graph to denote the causal relations among all the variables [11, 12, 13, 14, 15, 16]. Generally, we cannot identify all the causal relations from observational data, unless we make some additional functional assumptions [17, 18, 19] or exploit the abundant information in multiple or dynamic environments [20, 21].

In light of the uncertainty of the causal relations, a common practice to reveal them is introducing *background knowledge*, which is called *BK* for short. BK can be attained from *experiments* or *human expertise*. When experiments are available, we can collect interventional data to learn additional causal relations [22, 23, 24, 25, 26, 27, 28, 29, 30, 31]. And if in the causal discovery task, there are some variables familiar to humans, it is also possible that the human expertise can be helpful [32]. For example, if we study the causal relations among some variables including sales and prices, the causal relations such as price causes sales can be obtained directly according to human expertise.

When BK is available in addition to observational data, a fundamental problem is: *what causal relations are identifiable in the presence of latent variables*? This problem is fundamental for its implication on the maximally identifiable causal knowledge with the observational data and BK. Its difficulty results from the fact that, in addition to the BK itself, some other causal relations can also be learned when incorporating BK. For example, they can be identified on the basis of some restrictions, such as the causal relations are acyclic. It is quite challenging to find the *complete* characterization for such additional causal knowledge in the presence of latent variables, and the complete characterization is necessarily accompanied with theoretical guarantee for the existence of causal graphs consistent to the observational data and local BK that have *exactly different* causal relations for the unidentifiable ones. Unfortunately, the problem remains open.

In this paper, we solve the problem with *sound* and *complete* orientation rules when the background knowledge is given in a *local* form. In the presence of latent variables, a partial ancestral graph (PAG)

36th Conference on Neural Information Processing Systems (NeurIPS 2022).

can be learned by FCI algorithm from observational data [33, 34, 35]. PAG can imply the *existence* of causal relation between any two variables but not necessarily imply the causal *direction*. We say *BK is local*, if when the BK contains the causal information with respect to a variable $X$, for each variable adjacent to $X$ in the PAG, the BK implies whether $X$ causes it or not. The local BK is common in real tasks no matter it is from experiments or human expertise. For example, when we make experiments and collect the data under intervention on $X$, for each variable $V$ that has a causal relation with $X$, the interventional data can tell whether $X$ causes $V$; and businessman often has enough domain knowledge about price, thus they usually know whether price causes other variables or not, such as price causes sales and number of customers, and price is not caused by stocks. Given a PAG and local BK, we propose a set of orientation rules to determine some causal directions in the PAG. We prove that the rules are sound and complete, which state that all the causal relations that are identifiable given available information are *exactly* those determined by the proposed rules, thus closing the problem given local BK.

The establishment of orientation rules compatible with local BK makes causal discovery by interventions possible in the presence of latent variables. We propose the first general active learning framework for causal discovery, with the target of identifying a *maximal ancestral graph (MAG)*, which implies the causal relations when there are latent variables. Considering that intervention is expensive in reality, we hope to achieve the target with as few interventions as possible. Hence we present a baseline maximal entropy criterion, equipped with Metropolis-Hastings sampling, to select the intervention variable such that we can learn more causal relations by each intervention. Our contributions in this paper are twofold:

(1) We show what causal relations are identifiable given local background knowledge in the presence of latent confounders with sound and complete orientation rules.

(2) We give the first active learning framework for causal discovery that is applicable when latent variables exist, where maximal entropy criterion equipped with Metropolis-Hastings sampling is introduced to select intervention variables.

**Related works.** In the literature, Meek [36] established sound and complete rules, generally called *Meek rules*, for causal identification given BK under the causal sufficiency assumption. The assumption requires that there are no latent variables that cause more than one observed variable simultaneously. However, causal sufficiency is untestable in practice. When we apply causality in subjects such as biology, sociology, and economics, it is quite often that there are latent variables. For example, the macroeconomic policy influences purchase price, the population of customers, and advertising cost, but it is hard to evaluate it, thereby a latent confounder. Andrews et al. [37] showed that FCI algorithm is complete given *tiered* BK, where all variables are partitioned into disjoint sets with explicit causal order. While in many cases, e.g., when BK is revealed by interventions, BK is not tiered. And Jaber et al. [28] investigated the complete algorithm to learn a graph when there are additional interventional distribution, while such knowledge is not needed in our paper.

## 2 Preliminary

A graph $G = (\mathbf{V}, \mathbf{E})$ consists of a set of vertices $\mathbf{V} = \{V_1, \cdots, V_p\}$ and a set of edges $\mathbf{E}$. For any subset $\mathbf{V}' \subseteq \mathbf{V}$, the *subgraph induced by* $\mathbf{V}'$ is $G_{\mathbf{V}'} = (\mathbf{V}', \mathbf{E}_{\mathbf{V}'})$, where $\mathbf{E}_{\mathbf{V}'}$ is the set of edges in $\mathbf{E}$ whose both endpoints are in $\mathbf{V}'$. For a graph $G$, $\mathbf{V}(G)$ denotes the set of vertices in $G$. $G$ is a complete graph if there is an edge between any two vertices. The subgraph induced by an empty set is also a complete graph. $G[-\mathbf{V}']$ denotes the subgraph $G_{\mathbf{V}\setminus\mathbf{V}'}$ induced by $\mathbf{V}\setminus\mathbf{V}'$. Usually, bold letter (*e.g.*, $\mathbf{V}$) denotes a set of vertices and normal letter (*e.g.*, $V$) denotes a vertex. A graph is *chordal* if any cycle of length four or more has a chord, which is an edge joining two vertices that are not consecutive in the cycle. If $G = (\mathbf{V}, \mathbf{E})$ is chordal, the subgraph of $G$ induced by $\mathbf{V}' \subseteq \mathbf{V}$ is chordal.

A graph $G$ is *mixed* if the edges in $G$ are either directed $\rightarrow$ or bi-directed $\leftrightarrow$. The two ends of an edge are called *marks* and have two types *arrowhead* or *tail*. A graph is a *partial mixed graph (PMG)* if it contains directed edges, bi-directed edges, and edges with *circles* ($\circ$). The circle implies that the mark here could be either arrowhead or tail but is indefinite. $V_i$ is *adjacent* to $V_j$ in $G$ if there is an edge between $V_i$ and $V_j$. A path in a graph $G$ is a sequence of distinct vertices $\langle V_0, \cdots, V_n \rangle$ such that for $0 \leq i \leq n-1$, $V_i$ and $V_{i+1}$ are adjacent in $G$. An edge in the form of $V_i \circ\!\!-\!\!\circ V_j$ is a *circle edge*. The circle component in $G$ is the subgraph consisting of all the $\circ\!\!-\!\!\circ$ edges in $G$. Denote the set of vertices adjacent to $V_i$ in $G$ by $\mathrm{Adj}(V_i, G)$. A vertex $V_i$ is a

*parent* of a vertex $V_j$ if there is $V_i \to V_j$. A *directed path* from $V_i$ to $V_j$ is a path comprised of directed edges pointing to the direction of $V_j$. A *possible directed path* from $V_i$ to $V_j$ is a path without an arrowhead at the mark near $V_i$ on every edge in the path. $V_i$ is an *ancestor/possible ancestor* of $V_j$ if there is a *directed path/possible directed path* from $V_i$ to $V_j$ or $V_i = V_j$. $V_i$ is a *descendant/possible descendant* of $V_j$ if there is a *directed path/possible directed path* from $V_j$ to $V_i$ or $V_j = V_i$. Denote the set of *parent/ancestor/possible ancestor/descendant/possible descendant* of $V_i$ in $G$ by $\text{Pa}(V_i, G)/\text{Anc}(V_i, G)/\text{PossAn}(V_i, G)/\text{De}(V_i, G)/\text{PossDe}(V_i, G)$. If $V_i \in \text{Anc}(V_j, G)$ and $V_i \leftarrow V_j/V_i \leftrightarrow V_j$, it forms a *directed cycle/almost directed cycle*. $*$ is a wildcard that denotes any of the marks (arrowhead, tail, and circle). We make a convention that when an edge is in the form of $\circ\!-\!*$, the $*$ here cannot be a tail since in this case the circle can be replaced by an arrowhead due to the assumption of no selection bias.

A non-endpoint vertex $V_i$ is a collider on a path if the path contains $*\!\to V_i \leftarrow\!*$. A path $p$ from $V_i$ to $V_j$ is a *collider path* if $V_i$ and $V_j$ are adjacent or all the passing vertices are colliders on $p$. $p$ is a *minimal path* if there are no edges between any two non-consecutive vertices. A path $p$ from $V_i$ to $V_j$ is a *minimal collider path* if $p$ is a collider path and there is not a proper subset $\mathbf{V}'$ of the vertices in $p$ such that there is a collider path from $V_i$ to $V_j$ comprised of $\mathbf{V}'$. A triple $\langle V_i, V_j, V_k \rangle$ on a path is unshielded if $V_i$ and $V_k$ are not adjacent. $p$ is an *uncovered path* if every consecutive triple on $p$ is unshielded. A path $p$ is a *minimal possible directed path* if $p$ is minimal and possible directed.

A mixed graph is an *ancestral graph* if there is no directed or almost directed cycle (since we assume no selection bias, we do not consider undirected edges in this paper). An ancestral graph is a *maximal ancestral graph (MAG, denoted by $\mathcal{M}$)* if it is *maximal*, *i.e.*, for any two non-adjacent vertices, there is a set of vertices that *m-separates* them [33]. A path $p$ between $X$ and $Y$ in an ancestral graph $G$ is an *inducing path* if every non-endpoint vertex on $p$ is a collider and meanwhile an ancestor of either $X$ or $Y$. An ancestral graph is maximal if and only if there is no inducing path between any two non-adjacent vertices [33].

In an MAG, a path $p = \langle X, \cdots, W, V, Y \rangle$ is a *discriminating path for $V$* if $(1)$ $X$ and $Y$ are not adjacent, and $(2)$ every vertex between $X$ and $V$ on the path is a collider on $p$ and a parent of $Y$. Two MAGs are *Markov equivalent* if they share the same *m-separations*. A class comprised of all Markov equivalent MAGs is a *Markov equivalence class (MEC)*. We use a *partial ancestral graph (PAG, denoted by $\mathcal{P}$)* to denote an MEC, where a tail/arrowhead occurs if the corresponding mark is tail/arrowhead for all Markov equivalent MAGs, and a circle occurs otherwise.

For a PMG $\mathbb{M}$ that is obtained from a PAG $\mathcal{P}$ by orienting some circles to either arrowheads or tails, an MAG $\mathcal{M}$ is *consistent to the PMG $\mathbb{M}$* if $(1)$ the non-circle marks in $\mathbb{M}$ are also in $\mathcal{M}$, and $(2)$ $\mathcal{M}$ is in the MEC represented by $\mathcal{P}$. Sometimes we will omit the PAG $\mathcal{P}$ and just directly say a PMG $\mathbb{M}$ (obtained from the PAG $\mathcal{P}$) since in this paper we study the rules to incorporate local BK to a PAG. We say an MAG $\mathcal{M}$ *is consistent to the BK* if $\mathcal{M}$ is with the orientations dictated by the BK.

## 3 Sound and Complete Rules

In this section, we present the sound and complete orientation rules to orient a PAG $\mathcal{P}$ with local background knowledge (BK), where $\mathcal{P}$ is learned by observational data [11, 35] and $\mathbf{V}(\mathcal{P}) = \{V_1, V_2, \cdots, V_d\}$. The *local BK* regarding $X$ means that the BK directly implies and only directly implies all the true marks at $X$, denoted by $BK(X)$. We assume the absence of selection bias and that the BK is correct. The correctness indicates that there exists an MAG consistent to $\mathcal{P}$ and the BK. Without loss of generality, we suppose the local BK is regarding $V_1, V_2, \cdots, V_k, 1 \leq k \leq d$. That is, for any vertex $X \in V_1, V_2, \cdots, V_k$, all the marks at $X$ are known according to the local BK; and for any vertex $X \in V_{k+1}, \cdots, V_d$, the local BK does not directly imply any marks at $X$.

First, we show the orientation rules to incorporate local BK. They follow the rules of Zhang [35] for learning a PAG but with one replacement and one addition. Due to the page limit, we do not list them here but the replaced and additional ones. See Appendix A for the rules proposed by Zhang [35].

$\mathcal{R}'_4$: If $\langle K, \cdots, A, B, R \rangle$ is a discriminating path between $K$ and $R$ for $B$, and $B \circ\!-\!* R$, then orient $B \circ\!-\!* R$ as $B \to R$.

$\mathcal{R}_{11}$: If $A \to\!\!\!* B$, then $A \to B$.

---

**Algorithm 1:** Update a PMG with local background knowledge

---

**Input:** A PMG $\mathbb{M}_i$, $BK(X)$

**Output:** Updated graph $\mathbb{M}_{i+1}$

**1** For any $K \in \text{PossDe}(X, \mathbb{M}_i[-\mathbf{C}])$ and any $T \in \mathbf{C}$ such that $K \circ\!\!-\!\!* T$ in $\mathbb{M}_i$, orient $K \leftarrow\!\!* T$ (the mark at $T$ remains); for all $K \in \text{PossDe}(X, \mathbb{M}_i[-\mathbf{C}])$ such that $X \circ\!\!-\!\!* K$, orient $X \to K$;

**2** Orient the subgraph $\mathbb{M}_i[\text{PossDe}(X, \mathbb{M}_i[-\mathbf{C}])\backslash\{X\}]$ as follows until no feasible updates: for any two vertices $V_l$ and $V_j$ such that $V_l \circ\!\!-\!\!\circ V_j$, orient it as $V_l \to V_j$ if **(i)** $\mathcal{F}_{V_l}\backslash\mathcal{F}_{V_j} \neq \emptyset$ or **(ii)** $\mathcal{F}_{V_l} = \mathcal{F}_{V_j}$ as well as there is a vertex $V_m \in \text{PossDe}(X, \mathbb{M}_i[-\mathbf{C}])\backslash\{X\}$ not adjacent to $V_j$ such that $V_m \to V_l \circ\!\!-\!\!\circ V_j$;

**3** Apply the orientation rules until the graph is closed under the orientation rules.

---

Prop. 1 implies the soundness of $\mathcal{R}_4'$ to orient a PAG $\mathcal{P}$ or a PMG obtained from $\mathcal{P}$ with local BK. See Appendix A for the proof. $\mathcal{R}_{11}$ is immediate due to no selection bias assumption. In the following, we make a convention that when we say the orientation rules, they refer to $\mathcal{R}_1 - \mathcal{R}_3, \mathcal{R}_8 - \mathcal{R}_{10}$ of Zhang [35] and $\mathcal{R}_4', \mathcal{R}_{11}$. A PMG is *closed under the orientation rules* if the PMG cannot be oriented further by the orientation rules.

**Proposition 1.** *Given a PAG $\mathcal{P}$, for any PMG $\mathbb{M}$ that is obtained from $\mathcal{P}$ by orienting some circles in $\mathcal{P}$ (or $\mathbb{M} = \mathcal{P}$), $\mathcal{R}_4'$ is sound to orient $\mathbb{M}$ with local background knowledge.*

*Proof sketch:* If there is $B \leftarrow\!\!* R$ in an MAG consistent with the case of $\mathcal{R}_4'$, there must be a minimal collider path between $K$ and $R$ across $B$, in which case $B \leftarrow\!\!* R$ *should have been* identified in the PAG according to Zhao et al. [38], Zhang [35], contradiction. $\square$

Next, we will prove the completeness of the proposed orientation rules. It is somewhat complicated. We first give a roadmap for the proof process. There are mainly two parts. The first is that we present a complete algorithm to orient $\mathcal{P}$ with the local BK regarding $V_1, V_2, \cdots, V_k$. The second part is to prove that the algorithm orient the same marks as the proposed orientation rules. Combining these two parts, we conclude the orientation rules are complete to orient a PAG. The construction of the algorithm and the corresponding proof for the completeness of the algorithm in the first step is the most difficult part. To achieve the construction, we divide the whole process of orienting a PAG with BK regarding $V_1, V_2, \cdots, V_k$ into $k$ steps. Beginning from the PAG $\mathcal{P}$ ($\mathcal{P}$ is also denoted by $\mathbb{M}_0$), in the $(i+1)$-th ($0 \leq i \leq k-1$) step we obtain a PMG $\mathbb{M}_{i+1}$ from $\mathbb{M}_i$ by incorporating $BK(V_{i+1})$ and orienting some other circles further. To obtain the updated graph in each step, we propose an algorithm orienting a PMG with local BK incorporated in this step. Repeat this process by incorporating $BK(V_1), BK(V_2), \ldots, BK(V_k)$ sequentially, we obtain the PMG with incorporated BK regarding $V_1, \cdots, V_k$. We will prove that the $k$-step algorithm to orient PAG with local BK regarding $V_1, \cdots, V_k$ is complete, by an induction step that if the first $i$-step algorithm is complete to update the PAG $\mathcal{P}$ with BK regarding $V_1, \cdots, V_i$, then the $(i+1)$-step algorithm is complete to update $\mathcal{P}$ with BK regarding $V_1, \cdots, V_{i+1}$. Hence the proof in the first part completes. In the second part, we show that the $k$-step algorithm orients the same marks as the proposed orientation rules. We thus conclude that the orientation rules are sound and complete for causal identification in the presence of latent variables given local BK.

We present Alg. 1 to obtain $\mathbb{M}_{i+1}$ from $\mathbb{M}_i$ by incorporating $BK(V_{i+1})$. For brevity, we denote $V_{i+1}$ by $X$, and introduce a set of vertices $\mathbf{C}$ defined as $\mathbf{C} = \{V \in \mathbf{V}(\mathcal{P}) \mid V \ast\!\!\to X \in BK(X)\}$ to denote the vertices whose edges with $X$ will be oriented to ones with arrowheads at $X$ according to $BK(X)$ directly. In $\mathbb{M}_{i+1}$, there is $X \leftarrow\!\!* V$ for $V \in \mathbf{C}$ and $X \ast\!\!\to V$ for $V \in \{V \in \mathbf{V}(\mathcal{P}) \mid V \ast\!\!-\!\!\circ X \text{ in } \mathbb{M}_i\}\backslash\mathbf{C}$ oriented directly according to $BK(X)$. We define $\mathcal{F}_{V_l}^{\mathbb{M}_i} = \{V \in \mathbf{C} \cup \{X\} \mid V \ast\!\!-\!\!\circ V_l \text{ in } \mathbb{M}_i\}$ for any $V_l \in \text{PossDe}(X, \mathbb{M}_i[-\mathbf{C}])\backslash\{X\}$, which is denoted by $\mathcal{F}_{V_l}$ for short. $\mathcal{F}_{V_l}$ denotes the vertices in $\mathbf{C} \cup \{X\}$ whose edges with $V_l$ are oriented to ones with arrowheads at $V_l$ in the first step of Alg. 1.

In the first step of Alg. 1, the orientation at $X$ follows $BK(X)$, and the orientation at the vertices apart of $X$ is motivated as the necessary condition for the ancestral property. Speaking roughly, if there is an oriented edge $K \to T$ in the case of the first step, then no matter how we orient the other circles, there will be a directed or almost directed cycle, unless we introduce new unshielded colliders (which takes new conditional independences relative to those in $\mathcal{P}$), both of which are evidently

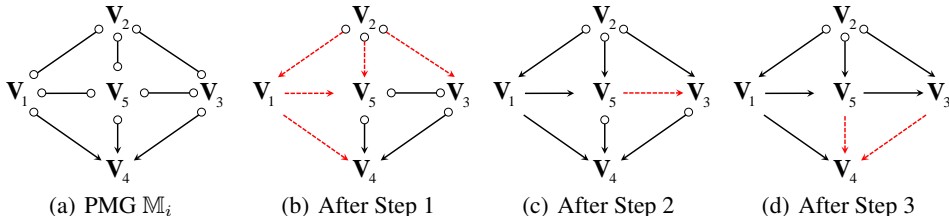

| (a) PMG $\mathbb{M}_i$ | (b) After Step 1 | (c) After Step 2 | (d) After Step 3 |

Figure 1: An example to demonstrate the implementation of each step of Alg. 1. Fig. 1(a) depicts a PMG $\mathbb{M}_i$. Suppose the local BK is in the form of $V_1 \leftarrow\!\ast V_2, V_1 \ast\!\rightarrow V_5, V_1 \ast\!\rightarrow V_4$. The Fig. 1(b)/1(c)/1(d) displays the graph obtained after the first/second/third step of Alg. 1. The edges oriented by each step are denoted by red dashed lines.

invalid to obtain an MAG in the MEC represented by $\mathcal{P}$. And the orientation in the second step is motivated as the necessary condition for that there are no new unshielded colliders in the oriented graph relative to the PAG $\mathcal{P}$. If there is an MAG where there is an inconsistent edge with the edge oriented in this step, then there must be new unshielded colliders relative to $\mathcal{P}$, which implies that the MAG is not consistent to $\mathcal{P}$. The third step orients some other circles based on the updated structure.

**Example 1.** *See the example in Fig. 1. Suppose the input PMG $\mathbb{M}_i$ in Alg. 1 is the graph shown in Fig. 1(a). And there is local BK regarding $X = V_1$, which is in the form of $V_1 \leftarrow\!\ast V_2, V_1 \ast\!\rightarrow V_5, V_1 \ast\!\rightarrow V_4$. Hence $\mathbf{C} = \{V_2\}$. In this case, $PossDe(X, \mathbb{M}_i[-\mathbf{C}]) = PossDe(V_1, \mathbb{M}_i[-V_2]) = \{V_1, V_3, V_4, V_5\}$. And $\mathcal{F}_{V_3} = \{V_2\}, \mathcal{F}_{V_4} = \{V_1\}, \mathcal{F}_{V_5} = \{V_1, V_2\}$. When we implement Alg. 1, in the first step, the edges denoted by red dashed lines in Fig. 1(b) are oriented. Among them, $V_1 \circ\!\!-\!\!\circ V_2/V_1 \circ\!\!-\!\!\circ V_5/V_1 \circ\!\!\rightarrow V_4$ is transformed to $V_1 \leftarrow\!\!\circ V_2/V_1 \rightarrow V_5/V_1 \rightarrow V_4$ due to $V_1 = X, V_2 \in \mathbf{C}, V_4, V_5 \in \{V \in \mathbf{V}(\mathcal{P}) \mid V \ast\!\!\circ X$ in $\mathbb{M}_i\} \backslash \mathbf{C}$; and $V_2 \circ\!\!\rightarrow V_5/V_2 \circ\!\!\rightarrow V_3$ is oriented due to $V_2 \in \mathbf{C}$ and $V_3, V_5 \in PossDe(X, \mathbb{M}_i[-\mathbf{C}])$. In the second step of Alg. 1, the edge denoted by red dashed line in Fig. 1(c) is oriented due to (1) a circle edge $V_3 \circ\!\!-\!\!\circ V_5$ after the first step, where $V_3, V_5 \in PossDe(X, \mathbb{M}_i[-\mathbf{C}])$; (2) $\mathcal{F}_{V_3} = \{V_2\} \subset \{V_1, V_2\} = \mathcal{F}_{V_5}$. In the third step of Alg. 1, the edges denoted by red dashed lines in Fig. 1(d) is oriented by $\mathcal{R}_1$ of the orientation rules.*

Then, we present the key induction result in Thm. 1 for the graph obtained by Alg. 1 in each step. Due to the page limit, we only show a proof sketch, with a detailed version in Appendix B. Then with Thm. 1, we directly conclude that $k$-step algorithm is complete to orient the PAG with the local BK regarding $V_1, \ldots, V_k$ in Cor. 1.

**Theorem 1.** *Given $i$, suppose $\mathbb{M}_s, \forall s \in \{0, 1, \ldots, i\}$ satisfies the five following properties:*

*(**Closed**) $\mathbb{M}_s$ is closed under the orientation rules.*

*(**Invariant**) The arrowheads and tails in $\mathbb{M}_s$ are invariant in all the MAGs consistent to $\mathcal{P}$ and BK regarding $V_1, \ldots, V_s$.*

*(**Chordal**) The circle component in $\mathbb{M}_s$ is chordal.*

*(**Balanced**) For any three vertices $A, B, C$ in $\mathbb{M}_s$, if $A \ast\!\!\rightarrow B \circ\!\!-\!\ast C$, then there is an edge between $A$ and $C$ with an arrowhead at $C$, namely, $A \ast\!\!\rightarrow C$. Furthermore, if the edge between $A$ and $B$ is $A \rightarrow B$, then the edge between $A$ and $C$ is either $A \rightarrow C$ or $A \circ\!\!\rightarrow C$ (i.e., it is not $A \leftrightarrow C$).*

*(**Complete**) For each circle at vertex $A$ on any edge $A \circ\!\!-\!\ast B$ in $\mathbb{M}_s$, there exist MAGs $\mathcal{M}_1$ and $\mathcal{M}_2$ consistent to $\mathcal{P}$ and BK regarding $V_1, \ldots, V_s$ with $A \leftarrow\!\ast B \in \mathbf{E}(\mathcal{M}_1)$ and $A \rightarrow B \in \mathbf{E}(\mathcal{M}_2)$.*

*Then the PMG $\mathbb{M}_{i+1}$ obtained from $\mathbb{M}_i$ with $BK(V_{i+1})$ by Alg. 1 also satisfies the five properties.*

*Proof sketch:* For brevity, we denote $V_{i+1}$ by $X$. **(A)** The closed property holds due to the third step of Alg. 1. **(B)** The invariant property holds because all the orientations in Alg. 1 either follow $BK(X)$ or are motivated as the necessary condition for the ancestral property and the fact that there cannot be new unshielded colliders introduced relative to $\mathbb{M}_i$. **(C)** The chordal property is proved based on the fact that only the first two steps of Alg. 1 possibly introduce new arrowheads, while the third step will only transform the edges as $A \circ\!\!\rightarrow B$ to $A \rightarrow B$, which is proved in Lemma 12 in Appendix B. With this fact, it suffices to prove that the circle component in the graph obtained after the first two steps is chordal. Denote the graph after the first two steps by $\bar{\mathbb{M}}_{i+1}$. We can prove that the circle components

in $\bar{\mathbb{M}}_{i+1}[\text{PossDe}(X, \mathbb{M}_i[-\mathbf{C}])]$ and in $\bar{\mathbb{M}}_{i+1}[-\text{PossDe}(X, \mathbb{M}_i[-\mathbf{C}])]$ are chordal, respectively. Since there are no circle edges connecting $\text{PossDe}(X, \mathbb{M}_i[-\mathbf{C}])$ and $\mathbf{V}\backslash\text{PossDe}(X, \mathbb{M}_i[-\mathbf{C}])$ (otherwise it has been oriented in the first step of Alg. 1), we conclude the desired result. **(D)** The balanced property of $\mathbb{M}_{i+1}$ is proved based on three facts that (1) in Alg. 1, if we transform a circle to arrowhead at $V$, then $V \in \text{PossDe}(X, \mathbb{M}_i[-\mathbf{C}])$; (2) if there is $A \in \text{PossDe}(X, \mathbb{M}_i[-\mathbf{C}])$ and $A \circ\!\!-\!\!* B$, $B \notin \mathbf{C}$, in $\mathbb{M}_{i+1}$, then $B \in \text{PossDe}(X, \mathbb{M}_i[-\mathbf{C}])$; (3) $\mathbb{M}_i$ satisfies the balanced property. We can prove that it is impossible that there is a sub-structure $V_i *\!\!\rightarrow V_j \circ\!\!-\!\!* V_k$ where $V_i$ is not adjacent to $V_k$ or there is $V_i *\!\!-\!\!\circ V_k$ in $\mathbb{M}_{i+1}$ by discussing whether $V_i, V_j, V_k$ belongs to $\text{PossDe}(X, \mathbb{M}_i[-\mathbf{C}])$. **(E)** The completeness property is proved by showing two results: (1) for edge circle edge $A \circ\!\!-\!\!\circ B$ and $C \circ\!\!\rightarrow D$ in $\mathbb{M}_{i+1}$, $C \circ\!\!\rightarrow D$ can be transformed to $C \rightarrow D$ and the circle edge can be oriented as both $A \rightarrow B$ and $A \leftarrow B$ in the MAGs consistent to $\mathcal{P}$ and local BK regarding $V_1, \cdots, V_{i+1}$; (2) in $\mathbb{M}_{i+1}$, each edge $A \circ\!\!\rightarrow B$ can be oriented as $A \leftrightarrow B$ in an MAG consistent to $\mathcal{P}$ and local BK regarding $V_1, \cdots, V_{i+1}$. In this part, the most difficult part is to prove the first result, with which the second result can be proved directly following the proof process of Thm. 3 of Zhang [35]. In the proof for the first result, we show that any MAG obtained from $\mathbb{M}_{i+1}$ by transforming the edges as $A \circ\!\!\rightarrow B$ to $A \rightarrow B$ and the circle component into a DAG without new unshielded colliders is consistent to $\mathcal{P}$ and local BK regarding $V_1, \ldots, V_{i+1}$. If not, we can always find an MAG obtained from $\mathbb{M}_i$ by transforming the edges as $A \circ\!\!\rightarrow B$ to $A \rightarrow B$ and the circle component into a DAG without new unshielded colliders that is not consistent to $\mathcal{P}$ and local BK regarding $V_1, \ldots, V_i$. By induction, there is an MAG obtained from $\mathcal{P}$ by transforming the edges as $A \circ\!\!\rightarrow B$ to $A \rightarrow B$ and the circle component into a DAG without new unshielded colliders that is not consistent to $\mathcal{P}$, contradiction with Thm. 2 of Zhang [35]. We conclude the first result. $\square$

**Corollary 1.** The $k$-step algorithm from $\mathbb{M}_0(=\mathcal{P})$ to $\mathbb{M}_k$ is sound and complete. That is, the non-circle marks in $\mathbb{M}_k$ are invariant in all the MAGs consistent to $\mathcal{P}$ and BK regarding $V_1, \ldots, V_k$. And for each circle in $\mathbb{M}_k$, there exist both MAGs with an arrowhead and MAGs with a tail here that are consistent to $\mathcal{P}$ and BK regarding $V_1, \ldots, V_k$.

*Proof.* Previous studies [34, 35] show that the last four properties in Thm. 1 are fulfilled in PAG, the case in $\mathcal{R}'_4$ will never happen in $\mathcal{P}$ because such circles have been oriented by $\mathcal{R}_4$ in the process of learning $\mathcal{P}$, and the case in $\mathcal{R}_{11}$ is never triggered by the rules to learn $\mathcal{P}$. Hence $\mathcal{P}$ satisfies the five properties. With the induction step implied by Thm 1, we directly conclude that $\mathbb{M}_k$ satisfies the five properties, thereby satisfying the invariant and complete property. $\square$

**Theorem 2.** *The orientation rules are sound and complete to orient a PAG with the local background knowledge regarding $V_1, \ldots, V_k$.*

*Proof.* The soundness of $\mathcal{R}'_4$ is shown by Prop. 1. The soundness of other rules immediately follows Thm. 4.1 of Ali et al. [34] and Thm. 1 of Zhang [35]. We do not show the details. Roughly speaking, the violation of these rules will lead to that there are new unshielded colliders or directed or almost directed cycles in the oriented graph relative to $\mathcal{P}$. The main part is to prove the completeness.

According to Cor. 1, it suffices to prove that in each step by Alg. 1 to incorporate $BK(X)$ into a PMG $\mathbb{M}_i$, the orientations in Alg. 1 either follow $BK(X)$ directly, or can be achieved by the proposed orientation rules. The orientation in the second step of Alg. 1 can be achieved by $\mathcal{R}_1$, because no matter $\mathcal{F}_{V_l}\backslash\mathcal{F}_{V_j} \neq \emptyset$ or $V_m \rightarrow V_l \circ\!\!-\!\!\circ V_j$, there is $F \in \mathcal{F}_{V_l}\backslash\mathcal{F}_{V_j}$ or $F = V_m$ respectively such that $F *\!\!\rightarrow V_l \circ\!\!-\!\!\circ V_j$ where $F$ is not adjacent to $V_j$. The orientation in the third step naturally follows the orientation rules. For the orientation in the first step, $X \leftarrow\!\!* V$ for $V \in \mathbf{C}$ is dictated by $BK(X)$, and $X \rightarrow V$ for $V \in \{V \in \mathbf{V}(\mathcal{P}) \mid X \circ\!\!\rightarrow V\}\backslash\mathbf{C}$ is obtained from $X \rightarrow\!\!* V$ dictated by $BK(X)$ and $\mathcal{R}_{11}$. The remaining part is to prove for $K \in \text{PossDe}(X, \mathbb{M}_i[-\mathbf{C}])\backslash\{X\}$ and $T \in \mathbf{C}$, if there is $K \circ\!\!-\!\!* T$ in $\mathbb{M}_i$, $K \leftarrow\!\!* T$ can be oriented by the proposed orientation rules when we incorporate $BK(X)$.

Due to $K \in \text{PossDe}(X, \mathbb{M}_i[-\mathbf{C}])\backslash\{X\}$, there is a possible directed path from $X$ to $K$ that does not go through $\mathbf{C}$. According to Lemma 2 in Appendix B, there is a minimal possible directed path $p = \langle X(=F_0), F_1, \ldots, K(=F_t)\rangle, t \geq 1$ where each vertex does not belong to $\mathbf{C}$. Hence $X \rightarrow F_1$ is oriented by $BK(X)$ and $\mathcal{R}_{11}$ unless $X \rightarrow F_1$ has been in $\mathbb{M}_i$. Hence, $X \rightarrow F_1 \rightarrow \cdots \rightarrow F_t$ can be oriented by $\mathcal{R}_1$ after incorporating $BK(X)$ unless they have been in $\mathbb{M}_i$. If $t = 1$, there is $T *\!\!\rightarrow X \rightarrow K$, thus $K \leftarrow\!\!* T$ can be oriented by $\mathcal{R}_2$. Next, we consider the case when $t \geq 2$.

We first prove that for any $F_m \in F_1, \ldots, F_t, t \geq 2$, $F_m$ is adjacent to $T$, and there is not $F_m \rightarrow T$ in $\mathbb{M}_i$. Suppose $F_m$ is not adjacent to $T$, there must be a sub-structure of $\mathbb{M}_i$ induced by

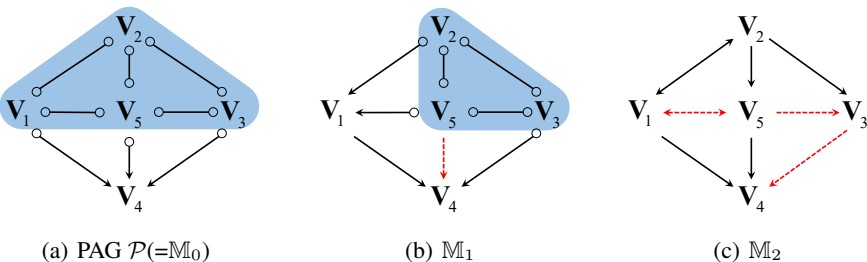



(a) PAG $\mathcal{P}(=\mathbb{M}_0)$      (b) $\mathbb{M}_1$      (c) $\mathbb{M}_2$



Figure 2: Fig. 2(a) depicts a PAG $\mathcal{P}$, with the local BK regarding $V_1$ in the form of $V_1 \leftarrow\!\ast V_2, V_1 \leftarrow\!\ast V_5, V_1 \ast\!\rightarrow V_4$ and the local BK regarding $V_2$ in the form of $V_2 \leftarrow\!\ast V_1, V_2 \ast\!\rightarrow V_3, V_2 \ast\!\rightarrow V_5$. The connected circle components are denoted by shaded area. The edges oriented by the orientation rules are denoted by red dashed lines.

$F_{m-s}, F_{m-s+1}, \ldots, F_{m+l}, T, 1 \leq s \leq m, 1 \leq l \leq t-m$, such that $T$ is only adjacent to $F_{m-s}$ and $F_{m+l}$ in this sub-structure. There are at least four vertices in this sub-structure. Hence there must be an unshielded collider (denoted by UC for short) in this sub-structure in $\mathcal{P}$, otherwise no matter how we orient the circle there is either a new UC relative to $\mathcal{P}$ or a directed or almost directed cycle there. Since $p$ is possibly directed, the UC is at either $F_{m+l}$ or $T$ (*i.e.*, $\ast\!\rightarrow F_{m+l}($ or $T) \leftarrow\!\ast)$. If there is a UC at $F_{m+l}$, $T\ast\!\rightarrow F_{m+l}$ and $F_{m+l-1}\ast\!\rightarrow F_{m+l}$ are identified in $\mathcal{P}$. Thus $F_{m+l} \rightarrow F_{m+l+1} \cdots \rightarrow F_t$ is identified in $\mathcal{P}$. Due to the completeness of FCI algorithm to learn $\mathcal{P}$, there is $K \leftrightarrow\!\ast T$ in $\mathcal{P}$, because there is not an MAG with $K \rightarrow T$ (there has been $T\ast\!\rightarrow F_{m+l} \rightarrow \cdots \rightarrow K$ in $\mathcal{P}$). Hence there is $K \leftrightarrow\!\ast T$ in $\mathbb{M}_i$, contradicting with $K \circ\!\ast T$ in $\mathbb{M}_i$. If there is not a UC at $F_{m+l}$, UC can only be at $T$. Thus $F_{m-s}\ast\!\rightarrow T \leftarrow\!\ast F_{m+l}$ is identified in $\mathcal{P}$. Since $p$ is possibly directed, $F_{m+l-1}$ is not adjacent to $T$, and there is not a UC at $F_{m+l}$ in the sub-structure, there cannot be $F_{m+l} \leftrightarrow T$ in $\mathcal{P}$. Hence the path $\langle F_{m-s}, F_{m-s+1}, \ldots, F_{m+l}, T \rangle$ in $\mathcal{P}$ is an uncovered possible directed path, $F_{m-s} \rightarrow T$ is identified in $\mathcal{P}$ (otherwise $\mathcal{R}_9$ applies). When incorporating $BK(X)$, there is a (almost) directed cycle $T\ast\!\rightarrow X \rightarrow \cdots \rightarrow F_{m-s} \rightarrow T$, contradicting with the correctness of BK. Hence, $F_m$ is adjacent to $T$. Similarly, if $F_m \rightarrow T$ in $\mathbb{M}_i$, there is $T\ast\!\rightarrow X \rightarrow \cdots \rightarrow F_m \rightarrow T$, impossibility.

Finally, since $F_1$ is adjacent to $T$, and $T\ast\!\rightarrow X \rightarrow F_1$ is oriented according to $BK(X)$, there is $T\ast\!\rightarrow F_1$ oriented by $\mathcal{R}_2$ unless $T\ast\!\rightarrow F_1$ has been in $\mathbb{M}_i$. Hence there is always $T\ast\!\rightarrow F_1$ by the orientation rules. Consider $T\ast\!\rightarrow F_1 \rightarrow F_2$, there is $T\ast\!\rightarrow F_2$ oriented by $\mathcal{R}_2$ unless $T\ast\!\rightarrow F_2$ has been in $\mathbb{M}_i$. Repeat the process for $F_3, F_4, \ldots, F_t(= K)$, we can prove that if there is $F_t(= K) \circ\!\ast T$ in $\mathbb{M}_i$, there is $T\ast\!\rightarrow F_t(= K)$ oriented by $\mathcal{R}_2$. The rules thus orient the same marks as Alg. 1. $\quad\square$

**Example 2.** *We give an example in Fig. 2. Suppose we obtain a PAG as Fig. 2(a) with observational data and have the local BK regarding $V_1, V_2$. We divide the whole process of obtaining a PMG from $\mathcal{P}$ with the local BK into obtaining $\mathbb{M}_1$ from $\mathcal{P}$ with $BK(V_1)$ by Alg. 1 and then obtaining $\mathbb{M}_2$ from $\mathbb{M}_1$ with $BK(V_2)$ by Alg. 1. $\mathbb{M}_1$ and $\mathbb{M}_2$ are shown in Fig. 2(b) and 2(c), respectively. It is not hard to verify that all of $\mathcal{P}, \mathbb{M}_1, \mathbb{M}_2$ satisfy the closed, chordal, and balanced properties defined in Thm 1. Note if we do not consider $\mathcal{R}'_4$, the edge colored red in Fig. 2(b) cannot be oriented. Fig. 2(a) also shows a case where $BK(V_1)$ is not tiered [37]. The reason is that the vertices $V_1, V_4, V_5$ cannot partitioned into disjoint subsets with explicit causal order because $V_1$ and $V_4$ belong to different subsets according to $BK(V_1)$ but $V_5$ has ancestor relation with neither $V_1$ nor $V_4$.*

## 4   Active Causal Discovery Framework

The establishment of the orientation rules for causal identification with local BK makes causal discovery by interventions possible in the presence of latent variables. Hence, on the basis of the theoretical results, we propose an active learning framework for causal discovery in the presence of latent variables, with the target of learning the MAG with as fewer interventions as possible. The framework is comprised of three stages. In Stage 1, we learn a PAG with observational data. In Stage 2, we select a singleton variable $X \in V_1, \ldots, V_d$ to intervene and collect the interventional data. In Stage 3, we learn causal relations with the data. For each edge $X \circ\!\ast V_i$, the circle at $X$ can be learned by a two-sample test on whether the interventional distribution of $V_i$ equals to the observational one. There is $X \leftarrow\!\ast V_i$ learned if they are equal, and $X \ast\!\rightarrow V_i$ otherwise. Hence, the knowledge taken by the

---
**Algorithm 2:** Intervention variable selection based on maximum entropy criterion with MH alg.
---
**Input:** A PMG $\mathbb{M}_i$ oriented based on $\mathcal{P}$ and BK regarding $V_1, \ldots, V_i$, number of MAGs $L$
**Output:** The selected intervention variable $X$

1   Obtain an MAG $\mathcal{M}_0$ based on $\mathbb{M}_i$ by transforming $\circ\!\!\rightarrow$ to $\rightarrow$ and the circle component into a DAG without new unshielded colliders;
2   **for** $t = 1, 2, \ldots, L'$ **do**
3     Sample an MAG $\mathcal{M}'$ from $S(\mathcal{M}_{t-1})$;
4     $\rho = \min(1, \frac{|S(\mathcal{M}_{t-1})|}{|S(\mathcal{M}')|})$;
5     Sample $u$ from uniform distribution $U[0,1]$;
6     **if** $u \leq \rho$ **then** $\mathcal{M}_t = \mathcal{M}'$ **else** $\mathcal{M}_t = \mathcal{M}_{t-1}$ ;
7   $\mathcal{S} = \{\mathcal{M}_{t,1 \leq t \leq L'} \mid \mathcal{M}_t \text{ has the non-circle marks in } \mathbb{M}_i\}$ $\triangleright$ The set of MAGs consistent to $\mathbb{M}_i$;
8   $s \leftarrow 0, X \leftarrow \emptyset$;
9   **for** $V_j = V_{i+1}, \ldots, V_d$ **do**
10    Denote $\mathbf{V}(V_j) = \{V \in \mathbf{V}(\mathbb{M}_i) \mid V_j \circ\!\!-\!\!* V \text{ in } \mathbb{M}_i\}$, $L = |\mathcal{S}|$;
11    For each possible local structure $\mathcal{L}_k$ of $V_j$, $1 \leq k \leq 2^{|\mathbf{V}(V_j)|}$, we count the number $\mathcal{N}_k$ of the appearance of $\mathcal{L}_k$ in the $L$ MAGs from $\mathcal{S}$;
12    $s' = -\sum_{k=1}^{2^{|\mathbf{V}(V_j)|}} \frac{\mathcal{N}_k}{L} \log \frac{\mathcal{N}_k}{L}$;
13    **if** $s \leq s'$ **then** $X \leftarrow V_j, s \leftarrow s'$;
14   **return** $X$.
---

interventional data is local. We repeat the second and third stages until we identify the MAG. Since the orientation rules are complete, the graph can be updated completely by each intervention. The only remaining problem is how to select the intervention variable in Stage 2.

Considering that the whole process is sequential, we only focus on the intervention variable selection in one round. Without loss of generality, suppose we have obtained a PMG $\mathbb{M}_i$ by $i$ interventions on $V_1, V_2, \ldots, V_i$, and will select a variable from $\{V_{i+1}, \ldots, V_d\}$ to intervene. We adopt the maximum entropy criterion [22]. For $\mathbb{M}_i$, we select the variable $X$ that maximizes

$$H_X = -\sum_{j=1}^{M} \frac{l_j}{L} \log \frac{l_j}{L}, \tag{1}$$

where $j$ is an index for a local structure of $X$ (a local structure of $X$ denotes a definite orientation of the marks at $X$), $M$ denotes the number of different local structures, $l_j$ denotes the number of MAGs consistent to $\mathbb{M}_i$ which has the $j$-th local structure of $X$, and $L$ denotes the total number of MAGs consistent to $\mathbb{M}_i$. Intuitively, the maximum entropy criterion is devoted to selecting the intervention variable $X$ such that there is a similar number of MAGs with each local structure of $X$ and as more as possible local structures of $X$. A justification for intervening on such a variable is that we hope to have a small space of MAGs after the intervention no matter what the true local structure of $X$ is.

However, it is hard to count the number of MAGs consistent to $\mathbb{M}_i$ with each definite local structure. Even in causal sufficiency setting, implementing such operation (generally called *counting maximally oriented partial DAGs*) is #P-complete [39]. Considering DAG is a special case for MAG, the counting of MAGs is harder. Hence, we adopt a sampling method based on Metropolis-Hastings (MH) algorithm [40], to uniformly sample from the space of MAGs. The algorithm begins from an MAG consistent to $\mathbb{M}_i$, and in each round we transform the MAG to a candidate MAG and decide to accept or reject it with some probability. Here, we introduce an important result of Zhang and Spirtes [41] for MAGs transformation in Prop. 2.

**Proposition 2** (Zhang and Spirtes [41], Tian [42]). *Let $\mathcal{M}$ be an arbitrary MAG, and $A \rightarrow B$ an arbitrary directed edge in $\mathcal{M}$. Let $\mathcal{M}'$ be the graph identical to $\mathcal{M}$ except that the edge between $A$ and $B$ is $A \leftrightarrow B$. $\mathcal{M}'$ is an MAG Markov equivalent to $\mathcal{M}$ if and only if*

> *(1) there is no directed path from $A$ to $B$ other than $A \rightarrow B$ in $\mathcal{M}$;*

> *(2) for any $C \rightarrow A$ in $\mathcal{M}$, $C \rightarrow B$ is also in $\mathcal{M}$; and for any $D \leftrightarrow A$ in $\mathcal{M}$, either $D \rightarrow B$ or $D \leftrightarrow B$ is in $\mathcal{M}$;*

> *(3) there is no discriminating path for $A$ on which $B$ is the endpoint adjacent to $A$ in $\mathcal{M}$.*

In the MAG sampling algorithm, in each step we transform the current MAG to a new MAG by *converting a directed edge to bi-directed edge or a bi-directed one to directed one*, where we use Prop. 2 to determine whether an MAG Markov equivalent to the current MAG can be obtained by the conversion. For MH algorithm, a stationary distribution equal to the desired distribution can be obtained if any two states can be transformed to each other in limited steps [43]. As implied by Theorem 3 of Zhang and Spirtes [41], any MAG can be transformed to another Markov equivalent MAG in a limited number of transformations above. Hence, MH algorithm is valid to sample MAGs uniformly from the space of MAGs consistent to $\mathcal{P}$. Then, we only remain the MAGs that have the same non-circle marks as $\mathbb{M}_i$. In this way, we obtain a set of MAGs which are uniformly sampled from the space of MAGs consistent to $\mathbb{M}_i$.

Given an MAG $\mathcal{M}$, let $S(\mathcal{M})$ denote the set of MAGs that can be obtained from $\mathcal{M}$ by transforming one bi-directed edge to directed edge or one directed edge to bi-directed edge according to Prop. 2. Denote the cardinality of $S(\mathcal{M})$ by $|S(\mathcal{M})|$. We set the probability $Q(\mathcal{M}' \mid \mathcal{M})$ of an MAG $\mathcal{M}$ transformed to another MAG $\mathcal{M}' \in S(\mathcal{M})$ as $1/|S(\mathcal{M})|$. Hence, the acceptance ratio $\rho$ that is used to decide whether to accept or reject the candidate is

$$\rho = \min\left(1, \frac{p(\mathcal{M}')Q(\mathcal{M} \mid \mathcal{M}')}{p(\mathcal{M})Q(\mathcal{M}' \mid \mathcal{M})}\right) = \min\left(1, \frac{|S(\mathcal{M})|}{|S(\mathcal{M}')|}\right).$$

We propose Alg. 2 to select the intervention variable $X$. As shown by Lemma 15.1 in Appendix B, the graph $\mathcal{M}_0$ is an MAG consistent to $\mathbb{M}_i$. From Line 2-Line 6, we execute MH algorithm to sample $L'$ MAGs. Then, we select the MAGs among them which are consistent to $\mathbb{M}_i$ on Line 7. Finally, we estimate the entropy by (1) and select $X$ from Line 9-Line 14.

## 5  Experiments

In this section, we conduct a simple simulation of the three-stage active learning framework. We generate 100 Erdös-Rényi random DAGs for each setting, where the number of variables $d = 10$ and the probability of including each edge $p \in \{0.1, 0.15, 0.2, 0.25, 0.3\}$. The weight of each edge is drawn from $\mathcal{U}[1, 2]$. We generate 10000 samples from the linear structural equations, and take three variables as latent variables and the others as observed ones. In the implementation of the MH algorithm in Alg. 2, we discard the first 500 sampled MAGs and collect the following 1000 MAGs. For each intervention variable $X$, we collect 10000 samples under $do(X = 2)$, and learn the circles at $X$ by two-sample test with a significance level of 0.05.

We compare the maximum entropy criterion with a baseline random criterion where we randomly select one variable with circles to intervene in each round. We show the results in Tab. 1. # int. denotes the number of interventions to achieve MAG identification. The effectiveness of the maximum entropy criterion is verified by noting that the number of interventions with maximum entropy criterion is fewer than that with random criterion. Further, we evaluate the three stages respectively. In Stage 1, we obtain a PAG by running FCI algorithm with a significance level of 0.05. In Stage 2, we adopt the two criteria to select intervention variables. In Stage 3, we learn the marks with corresponding interventional data and orientation rules. We evaluate the performance of Stage 1 by # correct PAG/# wrong PAG. # correct PAG/# wrong PAG denotes the number of edges that are correctly/wrongly identified by FCI. An edge is correctly/wrongly identified by FCI if the edge learned by FCI is identical/not identical to the true PAG. The performance of Stage 2 is evaluated by # int.. And we evaluate the performance of Stage 3 by # correct int./# wrong int., where # correct int./# wrong int. denotes the number of edges whose direction are correctly/wrongly identified by interventions. An edge is correctly/wrongly identified by interventions if its existence is correctly identified in $\mathcal{P}$ but the direction is uncertain, and after interventions we learn its direction correctly/wrongly. We evaluate the performance of the whole process by Norm. SHD and F1. Norm. SHD denotes the normalized structural hamming distance (SHD), which is calculated by dividing SHD by $d(d-1)/2$. F1 score is calculated by the confusion matrix to indicate whether the edge between any two vertices is correctly learned. According to the SHD and F1 score, the active framework can learn the MAG accurately when $p$ is not large. And as shown by the evaluations of Stage 1 and Stage 3, the marks are learned accurately in Stage 3, and most of the mistakes are generated in Stage 1. Hence, in the active learning framework, the PAG estimation in the first stage is the bottleneck of having a good performance.

Table 1: Number of interventions, normalized SHD, F1 score, number of correctly/ wrongly learned marks by interventions, and number of correctly/wrongly learned marks in PAG over 100 simulations with $d = 10$ and varying $p$ in the format of mean $\pm$ std.

| Strategy-$p$ | # int. | Norm. SHD | F1 | # correct int. | # wrong int. | # correct PAG | # wrong PAG |
|---|---|---|---|---|---|---|---|
| Random-0.10 | $2.88 \pm 1.28$ | $0.02 \pm 0.04$ | $0.85 \pm 0.29$ | $3.92 \pm 2.40$ | $0.11 \pm 0.53$ | $4.78 \pm 3.11$ | $0.39 \pm 0.82$ |
| MCMC-0.10 | $2.77 \pm 1.19$ | $0.02 \pm 0.04$ | $0.85 \pm 0.29$ | $4.00 \pm 2.40$ | $0.03 \pm 0.22$ | | |
| Random-0.15 | $3.30 \pm 1.15$ | $0.02 \pm 0.05$ | $0.91 \pm 0.17$ | $5.17 \pm 2.62$ | $0.10 \pm 0.41$ | $7.21 \pm 3.85$ | $0.40 \pm 0.92$ |
| MCMC-0.15 | $3.20 \pm 1.03$ | $0.02 \pm 0.04$ | $0.92 \pm 0.16$ | $5.25 \pm 2.66$ | $0.02 \pm 0.14$ | | |
| Random-0.20 | $3.59 \pm 1.22$ | $0.04 \pm 0.06$ | $0.91 \pm 0.15$ | $6.26 \pm 2.75$ | $0.19 \pm 0.61$ | $9.26 \pm 3.94$ | $0.59 \pm 1.30$ |
| MCMC-0.20 | $3.42 \pm 1.16$ | $0.03 \pm 0.06$ | $0.92 \pm 0.15$ | $6.38 \pm 2.70$ | $0.07 \pm 0.33$ | | |
| Random-0.25 | $3.47 \pm 1.34$ | $0.08 \pm 0.14$ | $0.89 \pm 0.18$ | $7.08 \pm 3.37$ | $0.06 \pm 0.34$ | $11.92 \pm 4.01$ | $1.59 \pm 2.85$ |
| MCMC-0.25 | $3.22 \pm 1.19$ | $0.08 \pm 0.14$ | $0.89 \pm 0.18$ | $7.05 \pm 3.39$ | $0.09 \pm 0.35$ | | |
| Random-0.30 | $3.64 \pm 1.32$ | $0.14 \pm 0.15$ | $0.83 \pm 0.18$ | $7.03 \pm 3.33$ | $0.36 \pm 1.01$ | $12.59 \pm 4.08$ | $2.63 \pm 3.19$ |
| MCMC-0.30 | $3.55 \pm 1.37$ | $0.14 \pm 0.15$ | $0.83 \pm 0.18$ | $7.25 \pm 3.50$ | $0.14 \pm 0.43$ | | |

# 6 Conclusion

In this paper, we show what causal relations are identifiable in the presence of latent variables given local background knowledge with sound and complete orientation rules. Based on the theoretical results, we give the first active learning framework for causal discovery in the presence of latent variables. In the future, we will investigate the causal relations identifiability with general background knowledge. It is also worthy to study how our research may help some recent novel decision-making methodology [44].

# Acknowledgment

This research was supported by NSFC (61921006), the Collaborative Innovation Center of Novel Software Technology and Industrialization, and the program A for Outstanding PhD candidate of Nanjing University. We are grateful to the reviewers for their valuable comments.

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
