# A    Orientation Rules for Causal Discovery with Observational Data

In this section, we show the complete orientation rules proposed by Zhang [35] for causal discovery with observational data in the presence of latent variables and selection bias. There are eleven rules ($\mathcal{R}_0 - \mathcal{R}_{11}$). Since selection bias is not considered in this paper, we do not show the cases ($\mathcal{R}_5 - \mathcal{R}_7$) that happen only when there is selection bias. $\mathcal{R}_0$ is triggered according to the conditional independence relationship at the beginning of learning a PAG. It is evidently not triggered after, hence we do not show it as well.

$\mathcal{R}_1$: If $A \mathbin{*}\!\!\to B \circ\!\!\mathbin{*} R$, and $A$ and $R$ are not adjacent, then orient the triple as $A \mathbin{*}\!\!\to B \to R$.

$\mathcal{R}_2$: If $A \to B \mathbin{*}\!\!\to R$ or $A \mathbin{*}\!\!\to B \to R$, and $A \mathbin{*}\!\!\circ R$, then orient $A \mathbin{*}\!\!\circ R$ as $A \mathbin{*}\!\!\to R$.

$\mathcal{R}_3$: If $A \mathbin{*}\!\!\to B \leftarrow\!\!\mathbin{*} R$, $A \mathbin{*}\!\!\circ D \circ\!\!\mathbin{*} R$, $A$ and $R$ are not adjacent, and $D \mathbin{*}\!\!\circ B$, then orient $D \mathbin{*}\!\!\circ B$ as $D \mathbin{*}\!\!\to B$.

$\mathcal{R}_4$: If $\langle K, \ldots, A, B, R \rangle$ is a discriminating path between $K$ and $R$ for $B$, and $B \circ\!\!\mathbin{*} R$; then if $B \in \text{Sepset}(K, R)$, orient $B \circ\!\!\mathbin{*} R$ as $B \to R$; otherwise orient the triple $\langle A, B, R \rangle$ as $A \leftrightarrow B \leftrightarrow R$.

$\mathcal{R}_8$: If $A \to B \to R$, and $A \circ\!\!\to R$, orient $A \circ\!\!\to R$ as $A \to R$.

$\mathcal{R}_9$: If $A \circ\!\!\to R$, and $p = \langle A, B, D, \ldots, R \rangle$ is an uncovered possible directed path from $A$ to $R$ such that $R$ and $B$ are not adjacent, then orient $A \circ\!\!\to R$ as $A \to R$.

$\mathcal{R}_{10}$: Suppose $A \circ\!\!\to R$, $B \to R \leftarrow D$, $p_1$ is an uncovered possible directed path from $A$ to $B$, and $p_2$ is an uncovered possible directed path from $A$ to $D$. Let $U$ be the vertex adjacent to $A$ on $p_1$ ($U$ could be $B$), and $W$ be the vertex adjacent to $A$ on $p_2$ ($W$ could be $D$). If $U$ and $W$ are distinct, and are not adjacent, then orient $A \circ\!\!\to R$ as $A \to R$.

In this paper, when we orient the PAG $\mathcal{P}$ with local BK, we replace $\mathcal{R}_4$ by $\mathcal{R}_4'$. We will show the soundness of $\mathcal{R}_4'$ in Prop. 1. Before that, we present a fact in Lemma 1.

**Lemma 1.** *If there exists a minimal collider path in an MAG $\mathcal{M}$ consistent to a PAG $\mathcal{P}$, then it is also a collider path in $\mathcal{P}$.*

*Proof.* Suppose a minimal collider path $p$ in $\mathcal{M}$, we consider its corresponding path in $\mathcal{P}$. If there exists a circle or tail at the non-endpoint vertex on this path, according to the completeness of FCI [35], there exists an MAG Markov equivalent to $\mathcal{M}$ that has a tail there, which contradicts Theorem 2.1 of Zhao et al. [38] that Markov equivalent MAGs have the same minimal collider paths. Hence the corresponding path of $p$ in $\mathcal{P}$ is also a collider path. □

**Proposition 1.** *Given a PAG $\mathcal{P}$, for any PMG $\mathbb{M}$ that is obtained from $\mathcal{P}$ by orienting some circles in $\mathcal{P}$ (or $\mathbb{M} = \mathcal{P}$), $\mathcal{R}_4'$ is sound to orient $\mathbb{M}$ with local background knowledge.*

*Proof.* Suppose there is a discriminating path $\langle K, \ldots, A, B, R \rangle$ between $K$ and $R$ for $B$, and $B \circ\!\!\mathbin{*} R$ in a PMG $\mathbb{M}$ such that there exists an MAG $\mathcal{M}$ consistent to $\mathbb{M}$. According to the definition of discriminating path and the soundness of $\mathcal{R}_2$, there is $B \circ\!\!\to R$. Suppose the violation of $\mathcal{R}_4'$, that is, in $\mathcal{M}$ there is $B \leftrightarrow R$. Since there is $A \to R$, the edge between $A$ and $B$ can only be $A \leftrightarrow B$ due to the ancestral property. Hence, there is a collider path between $K$ and $R$ as $K \mathbin{*}\!\!\to \cdots \leftrightarrow A \leftrightarrow B \leftrightarrow R$. If this collider path is a minimal one, then according to Lemma 1 the collider path is identifiable in $\mathcal{P}$, thus there is $A \leftrightarrow B \leftarrow\!\!\mathbin{*} R$ in $\mathbb{M}$, contradiction. Hence the collider path is not a minimal collider path from $K$ to $R$, there is a path $p_1$ comprised of a subset of these vertices that is a minimal collider path from $K$ to $R$. Note **(1)** for any vertex $V$ in the non-endpoint from $K$ to $B$, there is $V \to R$. And **(2)** $K$ is not adjacent to $R$. Hence the only vertex that can be adjacent to $R$ in $p_1$ is $B$. Hence the minimal path is as $\langle K, \ldots, B, R \rangle$. According to Lemma 1 again, $B \leftarrow\!\!\mathbin{*} R$ is identifiable in $\mathcal{P}$, thus there is $B \leftarrow\!\!\mathbin{*} R$ in $\mathbb{M}$, contradiction. We conclude the impossibility of the violation of $\mathcal{R}_4'$. □

# B    Proof of Theorem 1

For brevity, when we introduce a set of vertices $\mathbf{C}$ defined as $\mathbf{C} = \{V \in \mathbf{V}(\mathcal{P}) \mid V \mathbin{*}\!\!\to X \in BK(X)\}$ to denote the vertices whose edges with $X$ will be oriented to ones with arrowheads at $X$ according to $BK(X)$, we call the BK (of X) is *dictated by* $\mathbf{C}$.

We first show two definitions. A vertex $A$ of $G$ is called *simplicial* if its adjacency set $\text{Adj}(A, G)$ induces a complete subgraph of $G$. A *perfect elimination order* of a graph $G$ is an ordering $\sigma = (V_1, \ldots, V_n)$ of its vertices, so that each vertex $V_i$ is a simplicial vertex in the induced subgraph $G_{V_i, \ldots, V_n}$.

**Proposition 3** (Ali et al. [45], Zhang [35]). *In a PAG $\mathcal{P}$, for any three vertices $A, B, C$, if $A \ast\!\!\rightarrow B \circ\!\!\ast C$, then there is an edge between $A$ and $C$ with an arrowhead at $C$, namely, $A \ast\!\!\rightarrow C$. Furthermore, if the edge between $A$ and $B$ is $A \rightarrow B$, then the edge between $A$ and $C$ is either $A \rightarrow C$ or $A \circ\!\!\rightarrow C$ (i.e., it is not $A \leftrightarrow C$).*

**Proposition 4** (Spirtes and Richardson [46]). *Two MAGs over the same set of vertices are Markov equivalent if and only if*

*(1) They have the same adjacencies;*

*(2) They have the same unshielded colliders;*

*(3) If a path is a discriminating path for a vertex $V$ in both graphs, then $V$ is a collider on the path in one graph if and only if it is a collider on the path in the other.*

**Lemma 2.** *Consider $\mathbb{M}_i$ in Thm. 1 that satisfies the five properties. If there is a possible directed path from $A$ to $B$ in $\mathbb{M}_i$, then there is a minimal possible directed path from $A$ to $B$ in $\mathbb{M}_i$.*

*Proof.* If the path is minimal, then it trivially holds. If not, suppose the path comprised of $V_0(= A), V_1, \ldots, V_m(= B)$. As long as the path is not minimal, we can always find a sub-path comprised of $V_i, V_{i+1}, \ldots, V_j, j - i \geq 2$ such that any non-consecutive vertices in $V_i, \cdots, V_j$ are not adjacent except for an edge between $V_i$ and $V_j$. We will show that it is impossible that there is $V_i \leftrightarrow\!\!\ast V_j$ in $\mathbb{M}_i$. If $j - i = 2$, when there is an edge $V_i \leftrightarrow\!\!\ast V_j$ and an edge between $V_i$ and $V_{i+1}$ with a circle or tail at $V_i$, according to the balanced property and closed property of $\mathbb{M}_i$ under the orientation rules ($\mathcal{R}_2$ is triggered here) respectively, there is always an edge $V_{i+1} \leftrightarrow\!\!\ast V_j$, contradicting the possible directed path comprised of an edge from $V_{i+1}$ to $V_j$. If $j - i > 2$ and $V_i \leftrightarrow\!\!\ast V_j$, due to the non-adjacency of the vertices, there is either $V_i \rightarrow V_{i+1} \rightarrow \ldots V_j$ or $V_i \leftrightarrow\!\!\ast V_{i+1}$ identified in $\mathcal{P}$. The latter case is impossible due to the possible directed path. For the former case, there is an almost directed or directed cycles, contradiction. Hence, we can find a shorter possible directed path comprised of $V_0, V_1, \ldots, V_i, V_j, V_{j+1}, \ldots, V_m$ in $\mathbb{M}_i$. Repeat this process until we obtain a possible directed path that there is not a proper sub-structure where any non-consecutive vertices are not adjacent except for an edge between endpoints. This path is a minimal possible directed path. $\square$

**Lemma 3.** *Consider $\mathbb{M}_i$ in Thm. 1 that satisfies the five properties. If there is $A \ast\!\!\rightarrow B$ in $\mathbb{M}_i$, then there is an edge as $A \ast\!\!\rightarrow V$ for any $V$ in a connected circle component with $B$ in $\mathbb{M}_i$, and $A$ and $B$ are not in a connected circle component.*

*Proof.* It is a direct conclusion according to the balanced property of $\mathbb{M}_i$. We first consider any one vertex $V_1$ that is with a circle edge with $B$. That is, there is $A \ast\!\!\rightarrow B \circ\!\!-\!\!\circ V_1$ in $\mathbb{M}_i$. According to the balanced property of $\mathbb{M}_i$, there is an edge $A \ast\!\!\rightarrow V_1$. Similarly, we can conclude that the result holds for all the vertices in a connected circle component with $B$. Hence there cannot be a circle edge linking $A$ and any one vertex in a connected circle component with $B$. Thus $A$ and $B$ are not in a connected circle component. $\square$

**Lemma 4.** *Consider $\mathbb{M}_i$ in Thm. 1 that satisfies the five properties. Suppose an MAG $\mathcal{M}$ consistent to $\mathbb{M}_i$ and the local BK of $X$ dictated by $\mathbf{C}$. Then $V \in \text{PossDe}(X, \mathbb{M}_i[-\mathbf{C}])$ if and only if $V \in \text{De}(X, \mathcal{M})$.*

*Proof.* Suppose $\mathcal{M}$ an MAG consistent to $\mathbb{M}_i$ with the local BK of $X$ dictated by $\mathbf{C}$.

We prove the "only if" statement. If $V \in \text{PossDe}(X, \mathbb{M}_i[-\mathbf{C}])$, by Lemma 2, there is a minimal possible directed path $p$ from $X$ to $V$ comprised of $X, F_1, \ldots, F_m(= V)$. Due to $F_1 \notin \mathbf{C}$ and local BK, the edge between $X$ and $V_1$ can only be $X \rightarrow V_1$ in $\mathcal{M}$. Hence the path can only be directed in $\mathcal{M}$, otherwise there is at least one unshielded collider $F_{i-1} \ast\!\!\rightarrow F_i \leftrightarrow\!\!\ast F_{i+1}$ in $\mathcal{M}$, thus unshielded collider are identified in $\mathcal{P}$ and $\mathbb{M}_i$, contradicting with that $p$ is a minimal possible directed path from $X$ to $F_m$ in $\mathbb{M}_i$.

We then prove the "if" statement. There must be a minimal directed path $X \rightarrow F_1 \cdots \rightarrow F_{m-1}, F_m(= V)$ from $X$ to $V$ in $\mathcal{M}$. It is evident that $X$ cannot be adjacent to $F_2, \ldots, F_m$.

The corresponding path in $\mathbb{M}_i$ of this path is a minimal possible directed path from $X$ to $V$. If $V \notin \text{PossDe}(X, \mathbb{M}_i[-\mathbf{C}])$, there can only be $F_1 \in \mathbf{C}$ (since the vertices $F_2, F_3, \ldots, F_m$ are not adjacent to $X$). In this case $X \leftarrow\!*F_1$ should be dictated by $\mathbf{C}$ in $\mathcal{M}$, which contradicts the edge $X \rightarrow F_1$ in $\mathcal{M}$. The proof completes. $\qquad\square$

**Lemma 5.** *The PMG $\mathbb{M}_{i+1}$ in Thm. 1 satisfies the closed property.*

*Proof.* It is due to the third step of Alg. 1. $\qquad\square$

**Lemma 6.** *The PMG $\mathbb{M}_{i+1}$ in Thm. 1 satisfies the invariant property.*

*Proof.* We denote the oriented graph based on $\mathbb{M}_i$ and the local BK of $X$ dictated by $\mathbf{C}$ after the first two steps of Alg. 1 by $\bar{\mathbb{M}}_{i+1}$. Note in the third step of Algorithm 1 we just update the $\bar{\mathbb{M}}_{i+1}$ with the orientation rules. It is easy to prove the orientation rules are sound to orient $\bar{\mathbb{M}}_{i+1}$ referring to the results of Ali et al. [45], Zhang [35] because new unshielded colliders, or directed or almost directed cycles will be introduced otherwise, we thus do not present the details here. It suffices to show that the non-circle marks in $\bar{\mathbb{M}}_{i+1}$ are invariant in all MAGs consistent to $\mathbb{M}_i$ with the local BK of $X$ dictated by $\mathbf{C}$.

Consider $\forall K \in \text{PossDe}(X, \mathbb{M}_i[-\mathbf{C}])$ and $\forall T \in \mathbf{C}$. As shown by Lemma 4, for any MAG $\mathcal{M}$ consistent to $\mathbb{M}_i$ and the local BK dictated by $\mathbf{C}$, there is $K \in \text{De}(X, \mathcal{M})$. For brevity, in the following we call such MAG by *the MAG consistent to $\mathbb{M}_i$ and $\mathbf{C}$*. Considering $T\!*\!\rightarrow X \rightarrow \cdots \rightarrow K$, the edge between $K$ and $T$ can only be as $K \leftarrow\!*T$ in any MAG $\mathcal{M}$ if $K \neq X$, otherwise there is a directed or almost directed cycle in $\mathcal{M}$, contradiction. If $K = X$, the orientation $X \leftarrow\!*T$ in $\mathcal{M}$ just follows the local BK of $X$ dictated by $\mathbf{C}$.

Next we prove that the oriented edges in the second step are consistent to any MAG $\mathcal{M}$ consistent to $\mathbb{M}_i$ and $\mathbf{C}$. Suppose the edge between two vertices $V_j$ and $V_l$ oriented by the second step of Alg. 1 in $\mathbb{M}_i$ is not invariant in MAGs consistent to $\mathbb{M}_i$ and $\mathbf{C}$. That is, there is $V_l \leftarrow\!*V_j$ in an MAG $\mathcal{M}$ consistent to $\mathbb{M}_i$ and $\mathbf{C}$. The circle edges are oriented in two cases in the second step. We consider them one by one. **(A)** If $\mathcal{F}_{V_l} \backslash \mathcal{F}_{V_j} \neq \emptyset$ in $\mathbb{M}_i$, there exists some vertex $T \in \mathcal{F}_{V_l} \backslash \mathcal{F}_{V_j}$ forming a collider $V_j *\!\rightarrow V_l \leftarrow\!*T$ in $\mathcal{M}$. Then we prove the collider is unshielded. If $V_j$ is adjacent to $T$, we consider the edge in $\mathbb{M}_i$. **(a)** The edge is not $V_j \rightarrow T$, otherwise there must be a directed or almost directed cycles $X \rightarrow \cdots \rightarrow V_j \rightarrow T *\!\rightarrow X$ in $\mathcal{M}$; **(b)** the edge is not $V_j \circ\!\rightarrow T$, otherwise $T \in \mathcal{F}_{V_j}$; **(c)** the edge is not $V_j \leftarrow\!*T$, otherwise in $\mathbb{M}_i$ there is a sub-structure $T *\!\rightarrow V_j \circ\!\!-\!\!\circ V_l \circ\!*T$, contradicting with the balanced property of $\mathbb{M}_i$. Hence, $T$ cannot be adjacent to $V_j$. Thus $V_j *\!\rightarrow V_l \leftarrow\!*T$ is an unshielded collider. Thus $V_j *\!\rightarrow V_l$ is identifiable in $\mathcal{P}$. Since $\mathbb{M}_i$ is oriented based on $\mathcal{P}$, there is $V_j *\!\rightarrow V_l$ in $\mathbb{M}_i$, contradicting with $V_j \circ\!\!-\!\!\circ V_l$ in $\mathbb{M}_i$. **(B)** If there is $V_m \rightarrow V_j \circ\!\!-\!\!\circ V_l$ where $V_m \in \text{PossDe}(X, \mathbb{M}_i[-\mathbf{C}])\backslash\{X\}$ is not adjacent to $V_l$, there is an unshielded collider in $\mathcal{M}$ thus $V_l *\!\rightarrow V_i$ is identifiable in $\mathcal{P}$ and $\mathbb{M}_i$, contradiction. The proof completes. $\qquad\square$

**Lemma 7.** *Consider $\mathbb{M}_i$ in Thm. 1 that satisfies the five properties. For an edge $J \circ\!\!-\!\!\circ K$ satisfying $\mathcal{F}_J = \mathcal{F}_K$ in $\mathbb{M}_i[\text{PossDe}(X, \mathbb{M}_i[-\mathbf{C}])\backslash\{X\}]$, if it is oriented as $J \rightarrow K$ in the second step of Alg. 1 to obtain $\mathbb{M}_{i+1}$ based on $\mathbb{M}_i$ and $\mathbf{C}$, then there is a vertex $V_m \in \text{PossDe}(X, \mathbb{M}_i[-\mathbf{C}])\backslash\{X\}$ such that there is a minimal path $V_m \circ\!\!-\!\!\circ \ldots \circ\!\!-\!\!\circ V_1(= J) \circ\!\!-\!\!\circ V_0(= K), m \geq 1$ in $\mathbb{M}_i[\text{PossDe}(X, \mathbb{M}_i[-\mathbf{C}])\backslash\{X\}]$ where $\mathcal{F}_{V_m} \supset \mathcal{F}_{V_{m-1}} = \cdots = \mathcal{F}_{V_0}$.*

*Proof.* A directed edge $J \rightarrow K$ is oriented in the second step only if in two situations: (1) $\mathcal{F}_K \subset \mathcal{F}_J$; (2) $\mathcal{F}_K = \mathcal{F}_J$ and there is another vertex $L \in \text{PossDe}(X, \mathbb{M}_i[-\mathbf{C}])\backslash\{X\}$ that is not adjacent to $K$ and there is an edge $L \rightarrow J$ oriented in the second step (if $L \rightarrow J$ is not oriented in the second step, it can only be in $\mathbb{M}_i$. However, the $L \rightarrow J \circ\!\!-\!\!\circ K$ in $\mathbb{M}_i$ contradicts with the complete property of $\mathbb{M}_i$ because in this case there is $J \rightarrow K$ in any MAG consistent to $\mathbb{M}_i$). If $\mathcal{F}_{V_0} \subset \mathcal{F}_{V_1}$, in this case there is such a path for $m = 1$. If $\mathcal{F}_{V_0} = \mathcal{F}_{V_1}$, Then we could find some vertex $V_2 \in \text{PossDe}(X, \mathbb{M}_i[-\mathbf{C}])\backslash\{X\}$ that is not adjacent to $V_0$ and there is an edge $V_2 \rightarrow V_1$ oriented in the second step. And similar to the analysis for $V_1$, we conclude either $\mathcal{F}_{V_1} \subset \mathcal{F}_{V_2}$, in this case there is a path satisfying the result for $m = 2$; or $\mathcal{F}_{V_1} = \mathcal{F}_{V_2}$, in this case there is another vertex $V_3 \in \text{PossDe}(X, \mathbb{M}_i[-\mathbf{C}])\backslash\{X\}$ that is not adjacent to $V_1$ and there is an edge $V_3 \rightarrow V_2$ oriented. Repeat the process and we can always find an uncovered path $V_m \circ\!\!-\!\!\circ \ldots \circ\!\!-\!\!\circ V_1(= J) \circ\!\!-\!\!\circ V_0(= K), m \geq 1$ in $\mathbb{M}_i[\text{PossDe}(X, \mathbb{M}_i[-\mathbf{C}])\backslash\{X\}]$ where $\mathcal{F}_{V_0} = \cdots = \mathcal{F}_{V_{m-1}} \subset \mathcal{F}_{V_m}$. If the path is not minimal, then there exists a sub-structure $V_i \circ\!\!-\!\!\circ V_{i+1} \circ\!\!-\!\!\circ \cdots \circ\!\!-\!\!\circ V_j, j > i + 2$ where any two non-consecutive vertices are not adjacent except for an circle edge $V_i \circ\!\!-\!\!\circ V_j$ (the edge between $V_i$ and $V_j$ can only

be a circle edge, if it is $V_i \ast\to V_j$ or $V_i \leftrightarrow\ast V_j$, $V_i$ and $V_j$ cannot be in a connected circle component according to Lemma 3, but there is a circle path comprised of $V_i, V_{i+1}, \ldots, V_j$, contradiction). It contradicts with the fact that the circle component in $\mathbb{M}_i$ is chordal. Hence the path is minimal. $\quad\square$

**Lemma 8.** *Suppose $\mathbb{M}_s, 0 \leq s \leq i$ in Thm. 1 satisfy the five properties, there must exist an MAG consistent to $\mathbb{M}_i$.*

*Proof.* Suppose there does not exist an MAG consistent to $\mathbb{M}_i$. According to the invariant property of $\mathbb{M}_i$ and the basic assumption that the background knowledge is correct, there is not an MAG consistent to $\mathbb{M}_{i-1}$. Since $\mathbb{M}_s, 0 \leq s \leq i$ satisfies the invariant property, repeat the process above and we can conclude that there is not an MAG consistent to $\mathcal{P}$, contradiction. $\quad\square$

**Lemma 9.** *Consider $\mathbb{M}_{i+1}$ in Thm. 1. The subgraph of $\mathbb{M}_{i+1}$ induced by $\mathbf{C}$ is a complete graph.*

*Proof.* If it is not a complete graph, new unshielded colliders are introduced by the local background knowledge of $X$ dictated by $\mathbf{C}$ when obtaining $\mathbb{M}_{i+1}$ by Alg. 1. Hence there does not exist an MAG consistent to $\mathbb{M}_{i+1}$. According to the invariant property of $\mathbb{M}_{i+1}$ implied by Lemma 6 and the basic assumption that the background knowledge is correct, there is not an MAG consistent to $\mathbb{M}_i$, contradicting with Lemma 8. $\quad\square$

**Lemma 10.** *Suppose $\mathbb{M}_s, 0 \leq s \leq i$ in Thm. 1 satisfy the five properties. Then there is $\mathrm{PossDe}(X, \mathbb{M}_i[-\mathbf{C}]) \cap \mathrm{Pa}(\mathbf{C}, \mathbb{M}_i) = \emptyset$.*

*Proof.* Suppose there is an edge $V \to T$ where $V \in \mathrm{PossDe}(X, \mathbb{M}_i[-\mathbf{C}])$ and $T \in \mathbf{C}$ in $\mathbb{M}_i$. According to Lemma 4 and the definition of $\mathbf{C}$, for any graph oriented from $\mathbb{M}_i$ with the local background knowledge of $X$ dictated by $\mathbf{C}$, there will be a directed or almost directed cycle $X \to \cdots V \to T \ast\to X$. Hence there is not an MAG consistent to $\mathbb{M}_i$, contradicting with Lemma 8. $\quad\square$

**Lemma 11.** *Suppose $\mathbb{M}_s, 0 \leq s \leq i$ in Thm. 1 satisfy the five properties. In the second step of Alg. 1 to obtain $\mathbb{M}_{i+1}$ based on $\mathbb{M}_i$ and the local background knowledge of $X$ dictated by $\mathbf{C}$, there is not an edge oriented as both $J \leftarrow K$ and $J \to K$.*

*Proof.* For simplicity, we use $\mathbb{M}_i^1$ to denote $\mathbb{M}_i[\mathrm{PossDe}(X, \mathbb{M}_i[-\mathbf{C}])\backslash\{X\}]$. At first, we prove for any distinct vertices $J, K \in \mathbf{V}(\mathbb{M}_i^1)$, there is $\mathcal{F}_J \subseteq \mathcal{F}_K$ or $\mathcal{F}_K \subseteq \mathcal{F}_J$. Otherwise, there must exist at least two vertices $A, B \in \mathbf{C}$ such that there is $A \ast\!\!-\!\!\circ J$, $B \ast\!\!-\!\!\circ K$, where $A$ is not adjacent to $K$, and $B$ is not adjacent to $K$ in $\mathbb{M}_i^1$. Lemma 6 implies that the arrowhead added in the first step of Alg. 1 is invariant in all the MAGs consistent to $\mathbb{M}_i$ and local BK of $X$ dictated by $\mathbf{C}$ (we call such MAG by MAG consistent to $\mathbb{M}_i$ and $\mathbf{C}$ for short). Hence the added arrowheads in the first step appear in any MAG $\mathcal{M}$ consistent to $\mathbb{M}_i$ and $\mathbf{C}$. According to the condition, there are $A \ast\to J$ and $B \ast\to K$ in $\mathcal{M}$. In this case, there are always new unshielded colliders in $\mathcal{M}$ relative to $\mathbb{M}_i$ no matter what the orientation of the edge connecting $J$ and $K$ is in $\mathcal{M}$. Hence there are always new unshielded collider in the oriented graph relative to $\mathcal{P}$. That is, there does not exist an MAG consistent to $\mathbb{M}_i$ and $\mathbf{C}$. Due to the correctness of BK and Lemma 6, there is not an MAG consistent to $\mathbb{M}_i$, which contradicts with Lemma 8. Hence there is $\mathcal{F}_J \subseteq \mathcal{F}_K$ or $\mathcal{F}_K \subseteq \mathcal{F}_J$.

If $\mathcal{F}_J \neq \mathcal{F}_K$, without loss of generality, suppose $\mathcal{F}_J \subset \mathcal{F}_K$. Then $J \leftarrow K$ is oriented in the second step. If there is also $J \to K$ oriented in the second step, it implies there is $L \to J$ oriented in the second step where $L$ is not adjacent to $K$. In this case, no matter we orient $J \to K$ or $J \leftarrow K$, there is also a new unshielded collider at $J$ or $K$, hence there does not exist an MAG consistent to $\mathbb{M}_i$ and $\mathbf{C}$, a contradiction similar to the above case. In the following, we only consider the case of $\mathcal{F}_J = \mathcal{F}_K$. Suppose we orient both $J \to K$ and $J \leftarrow K$ in the second step.

By Lemma 7, if we orient $J \to K$ in the second step, there is a minimal circle path $V_0 \circ\!\!-\!\!\circ V_1 \circ\!\!-\!\!\circ \cdots \circ\!\!-\!\!\circ V_m(= J)$ where $\mathcal{F}_{V_m} \supset \mathcal{F}_{V_{m-1}} = \cdots = \mathcal{F}_{V_0}$. If we also orient $J \leftarrow K$ in the second step, there is a circle path $V_{m-1}(= J) \circ\!\!-\!\!\circ V_m(= K) \circ\!\!-\!\!\circ \cdots \circ\!\!-\!\!\circ V_n, n > m$ in $\mathbb{M}_i^1$ where $\mathcal{F}_{V_{m-1}} = \mathcal{F}_{V_m} = \cdots = \mathcal{F}_{V_{n-1}} \subset \mathcal{F}_n$. Note $V_{m+1}$ is adjacent to $V_m$ but is not adjacent to $V_{m-1}$, while $V_{m-2}$ is adjacent to $V_{m-1}$ but not adjacent to $V_m$, hence $V_{m-2} \neq V_{m+1}$, and $V_{m-2}, V_{m-1}, V_m, V_{m+1}$ are distinct vertices. Also note no circle edges in $\mathbb{M}_i^1$ are oriented in the first step of Alg. 1. Hence the circle component in $\mathbb{M}_i^1$ is still chordal. Hence $V_0 \circ\!\!-\!\!\circ V_1 \circ\!\!-\!\!\circ \cdots \circ\!\!-\!\!\circ V_n$ is also a minimal circle path, otherwise there must be a cycle comprised of circle edges whose length is larger than 3 without a chord because this cycle must contain $V_{m-2}, V_{m-1}, V_m, V_{m+1}$ where $V_{m-2}$ is not adjacent

to $V_m$ and $V_{m-1}$ is not adjacent to $V_{m+1}$, contradiction. Hence we consider the minimal circle path $V_0 \circ\!\!-\!\!\circ V_1 \circ\!\!-\!\!\circ \cdots \circ\!\!-\!\!\circ V_n$. According to Lemma 6, there must be $V_0 \to \cdots \to V_{m-1}$ and $V_m \leftarrow \cdots V_n$ in any MAG $\mathcal{M}$ consistent to $\mathbb{M}_i$ and $\mathbf{C}$. However, in this case there are new unshielded colliders in $\mathcal{M}$ relative to $\mathbb{M}_i$ and $\mathbf{C}$ no matter what the orientation of the edge connecting $V_{m-1}$ and $V_m$ is, that is, $\mathcal{M}$ is always not consistent to $\mathcal{P}$. Hence there does not exist an MAG consistent to $\mathbb{M}_i$ and $\mathbf{C}$, a contradiction similar to the above case. The proof completes. $\qquad\square$

**Lemma 12.** *Consider $\mathbb{M}_i$ in Thm. 1 that satisfies the five properties. In the third step of Alg. 1 to obtain $\mathbb{M}_{i+1}$ based on $\mathbb{M}_i$ and the local BK of $X$ dictated by $\mathbf{C}$, there are only edges as $A \leftarrow\!\!\circ B$ transformed to $A \leftarrow B$.*

*Proof.* There are three possible transformations by the orientation rules: there are edges as $A \circ\!\!-\!\!\circ B$ transformed to the edges with arrowheads; there are edges as $A \leftarrow\!\!\circ B$ transformed to $A \leftrightarrow B$; there are edges as $A \leftarrow\!\!\circ B$ transformed to $A \leftarrow B$ ($A \circ\!\!\to B$ is equivalent to $A \leftarrow\!\!\circ B$ due to the generality of $A$ and $B$). We will prove the impossibility of the first two cases. We denote the graph obtained from $\mathbb{M}_i$ and BK regarding $V_{i+1}$ after the first two steps of Alg. 1 by $\bar{\mathbb{M}}_{i+1}$. The proof idea is, suppose in the third step we orient some edges as $A \leftrightarrow R$ or orient some circle edges. We can always find the first edge which is transformed from $A \leftarrow\!\!\circ R$ to $A \leftrightarrow R$ or from circle edges to some directed or bi-directed edges in the third step. If we prove that this edge can be neither an edge as $A \leftarrow\!\!\circ R$ transformed to $A \leftrightarrow B$ nor a circle edge, we have a contradiction. Hence we can conclude that there are no edges as $A \leftarrow\!\!\circ R$ transformed to $A \leftrightarrow R$ or circle edges transformed to directed or bi-directed edges in the third step.

**If we orient some edges $A \leftarrow\!\!\circ R$ as $A \leftrightarrow R$ or orient some circle edges in the third step, the first such edge is not as $A \leftarrow\!\!\circ R$ and transformed to $A \leftrightarrow R$.** Suppose $A \leftarrow\!\!\circ R$ is transformed to $A \leftrightarrow R$ by the third step of Alg. 1. Note that an arrowhead is introduced by the orientation rules. We analyze the orientation rules. $\mathcal{R}_3$ is triggered in only the process of obtaining $\mathcal{P}$. $\mathcal{R}_4'$ does not transform an edge as $A \leftarrow\!\!\circ R$ to a bi-directed edge. $\mathcal{R}_8 - \mathcal{R}_{10}$ only introduces tails. Hence only $\mathcal{R}_1$ and $\mathcal{R}_2$ possibly introduce arrowheads. $\mathcal{R}_1$ cannot transform an edge $A \leftarrow\!\!\circ R$ to $A \leftrightarrow R$. Hence it suffices to prove there are no edges $A \leftarrow\!\!\circ R$ transformed to $A \leftrightarrow R$ by $\mathcal{R}_2$ in the third step of Alg. 1. According to the condition of $\mathcal{R}_2$, when $A \leftarrow\!\!\circ R$ is transformed to $A \leftrightarrow R$ by $\mathcal{R}_2$, there is **(i)** $A \to B \leftrightarrow R \circ\!\!\to A$ or **(ii)** $A \leftrightarrow B \to R \circ\!\!\to A$. We then prove two results: (**1**) the bi-directed edges in **(i)** or **(ii)** cannot appear in $\mathbb{M}_i$. (**2**) the bi-directed edges cannot be introduced in the first two steps of Alg. 1 to obtain $\mathbb{M}_{i+1}$ based on the local BK of $X$ dictated by $\mathbf{C}$.

(**1**) For **(i)**, suppose there is $B \leftrightarrow R$ in $\mathbb{M}_i$. Since after the first two steps there is $A \leftarrow\!\!\circ R$, there is $A *\!\!-\!\!\circ R$ in $\mathbb{M}_i$. According to the balanced property of $\mathbb{M}_i$, there is $A \leftarrow\!\!*B$ in $\mathbb{M}_i$, in which case there cannot be an edge $A \to B$ as **(i)**. For **(ii)**, suppose there is $A \leftrightarrow B$ in $\mathbb{M}_i$. Since after the first two steps there is $B \to R$, there must be $B \to R$ or $B \circ\!\!-\!\!* R$ in $\mathbb{M}_i$. For the former case, $A *\!\!\to R$ is oriented in $\mathbb{M}_i$ since $\mathbb{M}_i$ is closed under $\mathcal{R}_2$. For the latter case, $A *\!\!\to R$ is oriented in $\mathbb{M}_i$ due to the balanced property of $\mathbb{M}_i$. Both of them contradict with $A *\!\!-\!\!\circ R$ in the graph after the first two steps.

(**2**) For **(i)**, there is $R \circ\!\!-\!\!* A$ in $\mathbb{M}_i$. If $B \leftrightarrow R$ is oriented in the first two steps of Alg. 1, there is either $B \circ\!\!\to R$ or $B \leftarrow\!\!\circ R$ in $\mathbb{M}_i$. For the former case, according to the balanced property there is $A \leftarrow\!\!*B$ in $\mathbb{M}_i$ due to $R \circ\!\!-\!\!* A$, which contradicts with the structure $A \to B$ in **(i)**. For the latter case, since $B \leftarrow\!\!\circ R$ is transformed to $B \leftrightarrow R$ by the first two steps of Alg. 1, there is $R \in \mathrm{PossDe}(X, \mathbb{M}_i[-\mathbf{C}])$ and $B \in \mathbf{C}$. We discuss whether $A \in \mathbf{C}$, if $A \in \mathbf{C}$, there is $A *\!\!\to R$ oriented by the first step of Alg. 1, contradicting with the structure in **(i)**; if $A \notin \mathbf{C}$, since there is $A *\!\!-\!\!\circ R$ after the first two steps, there is $A *\!\!-\!\!\circ R$ in $\mathbb{M}_i$, there is $A \in \mathrm{PossDe}(X, \mathbb{M}_i[-\mathbf{C}])$, hence $A \leftarrow\!\!*B$ is oriented in the first step, contradicting with the structure in **(i)**. The contradiction for **(ii)** is similar, we thus do not present the details.

Combining the results in (**1**) and (**2**), for the first edge in the third step that is transformed from $A \leftarrow\!\!\circ R$ to $A \leftrightarrow R$ or transformed from a circle edge to a directed or bi-directed edge, the edge cannot be transformed by $\mathcal{R}_2$ as well. Hence the first edge mentioned above is not an edge transformed from $A \leftarrow\!\!\circ R$ to $A \leftrightarrow R$ because no orientation rules can achieve it.

**If we orient some edges $A \leftarrow\!\!\circ R$ as $A \leftrightarrow R$ or orient some circle edges in the third step, the first such edge is not a circle edge.** We prove that the first edge in the third step that is transformed from $A \leftarrow\!\!\circ R$ to $A \leftrightarrow R$ or transformed from a circle edge to a directed or bi-directed edge cannot be a circle edge. We analyze the orientation rules respectively. The result is evident for $\mathcal{R}_8 - \mathcal{R}_{10}$ since the transformed edge is as $A \circ\!\!\to R$, which is not a circle edge. $\mathcal{R}_3$ is triggered in only the process

of obtaining $\mathcal{P}$. When an edge is oriented by $\mathcal{R}_4'$, it can be seen as that we first transform a circle to an arrowhead by $\mathcal{R}_2$, then transform the other circle to tail by $\mathcal{R}_4'$. Hence it suffices to show that there are no circle edge oriented by $\mathcal{R}_1$ and $\mathcal{R}_2$ in the third step of Alg. 1. We first consider $\mathcal{R}_1$. Suppose there is $A *\!\!\rightarrow B \circ\!\!-\!\!\circ R$ where $A$ and $R$ are not adjacent after the first two steps of updating $\mathbb{M}_i$ with the local BK of $X$ dictated by $\mathbf{C}$. Since $\mathbb{M}_i$ satisfies the complete property, the arrowhead at $B$ on the edge $A *\!\!\rightarrow B$ can only be oriented in the first two steps, otherwise the arrowhead is in $\mathbb{M}_i$ and there is either $B \rightarrow R$ or $B \leftarrow\!\!* R$ in $\mathbb{M}_i$. Note the fact that in the first two steps of Alg. 1 we only add arrowheads at the vertex in $\mathrm{PossDe}(X, \mathbb{M}_i[-\mathbf{C}])$. Hence $B \in \mathrm{PossDe}(X, \mathbb{M}_i[-\mathbf{C}])$. In addition, there is $R \notin \mathbf{C}$, otherwise $B \leftarrow\!\!\circ R$ is oriented in the first step of Alg. 1. Hence there is $R \in \mathrm{PossDe}(X, \mathbb{M}_i[-\mathbf{C}])$ by Lemma 2. The edge $A *\!\!\rightarrow B$ is oriented by either the first or the second step. If $A *\!\!\rightarrow B$ is oriented by the first step, then $B \rightarrow R$ should be oriented in the second step since $A \in \mathcal{F}_B \backslash \mathcal{F}_R$; if $A *\!\!\rightarrow B$ is oriented by the second step, then $B \rightarrow R$ is also oriented by the second step, in both of cases there is not $B \circ\!\!-\!\!\circ R$ after the first two steps. Hence $\mathcal{R}_1$ is not triggered in the third step.

Then we consider that a circle edge is oriented by $\mathcal{R}_2$. Suppose there is $A \rightarrow B *\!\!\rightarrow R$ or $A *\!\!\rightarrow B \rightarrow R$, and $A \circ\!\!-\!\!\circ R$. We consider the cases: (i) the arrowhead at $R$ on the edge connecting $B$ and $R$ appears in $\mathbb{M}_i$; (ii) the arrowhead at $R$ on the edge connecting $B$ and $R$ is introduced by the first two steps of Alg. 1 to obtain $\mathbb{M}_{i+1}$ based on $\mathbb{M}_i$ and the local BK of $X$ dictated by $\mathbf{C}$.

(i) For the first case, there is $B *\!\!\rightarrow R$ and $A \circ\!\!-\!\!\circ R$ in $\mathbb{M}_i$. According to the balanced property of $\mathbb{M}_i$, there is $A \leftarrow\!\!* B$ in $\mathbb{M}_i$. Hence the only case that $\mathcal{R}_2$ is triggered is that there is $A \leftrightarrow B \rightarrow R \circ\!\!-\!\!\circ A$ after the first two steps, in which case there can only be $A \leftarrow\!\!\circ B$ in $\mathbb{M}_i$ due to the balance property. In this case, $A \leftarrow\!\!\circ B$ is transformed to $A \leftrightarrow B$ in only the first step. It implies that $A \in \mathbf{C}$ and $B \in \mathrm{PossDe}(X, \mathbb{M}_i[-\mathbf{C}])$. If $R \in \mathbf{C}$, there is $B \leftrightarrow R$ oriented in the first step, contradicting with $B \rightarrow R$ after the first two steps. If $R \notin \mathbf{C}$, since there is $B \in \mathrm{PossDe}(X, \mathbb{M}_i[-\mathbf{C}])$ and $B \rightarrow R$ or $B \circ\!\!-\!\!* R$ in $\mathbb{M}_i$, there is $B \in \mathrm{PossDe}(X, \mathbb{M}_i[-\mathbf{C}])$, thus there is $A *\!\!\rightarrow R$ oriented in the first step, contradicting with $A \circ\!\!-\!\!\circ R$ after the first two steps. Hence case (i) is impossible.

(ii) For the second case, note in the first two steps of Alg. 1 we only add arrowheads at the vertex in $\mathrm{PossDe}(X, \mathbb{M}_i[-\mathbf{C}])$, there is thus $R \in \mathrm{PossDe}(X, \mathbb{M}_i[-\mathbf{C}])$. In this case there is $A \notin \mathbf{C}$, otherwise $A *\!\!\rightarrow R$ is oriented by the first step, contradiction. Due to $R \in \mathrm{PossDe}(X, \mathbb{M}_i[-\mathbf{C}])$ and $A \circ\!\!-\!\!\circ R$ in $\mathbb{M}_i$, there is $A \in \mathrm{PossDe}(X, \mathbb{M}_i[-\mathbf{C}])$ according to Lemma 2. We discuss whether $B \in \mathbf{C}$. (ii.1) If $B \in \mathbf{C}$, the only case that $\mathcal{R}_2$ is triggered is that $A \leftrightarrow B \rightarrow R$ in $\bar{\mathbb{M}}_{i+1}$, which implies that there is $A *\!\!\rightarrow B$ and $B \rightarrow R$ or $B \circ\!\!-\!\!* R$ in $\mathbb{M}_i$. If $B \rightarrow R$ in $\mathbb{M}_i$, according to the closed property of $\mathbb{M}_i$ under $\mathcal{R}_1$, there is $A *\!\!\rightarrow R$ in $\mathbb{M}_i$, thus there is $A *\!\!\rightarrow R$ in $\bar{\mathbb{M}}_{i+1}$, contradiction. If $A *\!\!\rightarrow B \circ\!\!-\!\!* R$ in $\mathbb{M}_i$, according to the balanced property of $\mathbb{M}_i$, there is also $A *\!\!\rightarrow R$ in $\mathbb{M}_i$, thus there is $A *\!\!\rightarrow R$ in $\bar{\mathbb{M}}_{i+1}$, contradiction. (ii.2) If $B \notin \mathbf{C}$, if there exists an edge between $A, B, R$ that is not a circle edge in $\mathbb{M}_i$, due to the balanced property of $\mathbb{M}_i$ and $A \circ\!\!-\!\!\circ R$ in $\mathbb{M}_i$, there can be either $A *\!\!\rightarrow B \leftarrow\!\!* R$ or $A \leftarrow\!\!* B *\!\!\rightarrow R$ in $\mathbb{M}_i$. We just show the proof for the first case, and that for the other one is similar. If the case in $\mathcal{R}_2$ happens, there can only be $A \rightarrow B \leftrightarrow R$ in $\bar{\mathbb{M}}_{i+1}$. Since we never add a new bi-directed edge between $\mathrm{PossDe}(X, \mathbb{M}_i[-\mathbf{C}])$ in Alg. 1, the edge $B \leftrightarrow R$ is in $\mathbb{M}_i$. However, in this case due to balanced property of $\mathbb{M}_i$ and $A \circ\!\!-\!\!\circ R$ in $\mathbb{M}_i$, there is $A \leftrightarrow B$ in $\mathbb{M}_i$, contradicting with $A \rightarrow B$ in $\bar{\mathbb{M}}_{i+1}$. Hence in $\mathbb{M}_i$ there can only be $A \circ\!\!-\!\!\circ B \circ\!\!-\!\!\circ R \circ\!\!-\!\!\circ A$. Note the edge between $\mathrm{PossDe}(X, \mathcal{P}[-\mathbf{C}])$ is oriented in only the second step of Alg. 1, where we transform circle edges to directed edges, hence there is $A \rightarrow B \rightarrow R$ in $\bar{\mathbb{M}}_{i+1}$. Then we will prove the impossibility of $A \rightarrow B \rightarrow R \circ\!\!-\!\!\circ A$ in $\bar{\mathbb{M}}_{i+1}$. According to Lemma 7 and Lemma 11, if $A \rightarrow B \rightarrow R$ is oriented, then there is $\mathcal{F}_A \supseteq \mathcal{F}_B \supseteq \mathcal{F}_R$. If there is $\mathcal{F}_A \supset \mathcal{F}_B$ or $\mathcal{F}_B \supset \mathcal{F}_R$, then there is $\mathcal{F}_A \supset \mathcal{F}_R$, hence there is $A \rightarrow R$ oriented by the second step of Alg. 1, contradiction. If there is $\mathcal{F}_A = \mathcal{F}_B = \mathcal{F}_R$, we will prove its impossibility. According to Alg. 1, there is another vertex $C \in \mathrm{PossDe}(X, \mathbb{M}_i[-\mathbf{C}])$ such that $C \rightarrow A$ is oriented by the second step of Alg. 1, $C$ is not adjacent to $B$, and $\mathcal{F}_C \supseteq \mathcal{F}_A$. Hence there is $\mathcal{F}_C \supseteq \mathcal{F}_R$. We can see that $R$ must be adjacent to $C$, otherwise $A \rightarrow R$ will be oriented to $A \rightarrow R$ in the second step of Alg. 1. Due to Lemma 6, in each MAG $\mathcal{M}$ consistent to $\mathbb{M}_i$ and $\mathbf{C}$, there is $C \rightarrow A \rightarrow B \rightarrow R$, hence there can only be $C \rightarrow R$ in $\mathcal{M}$. In this case there is a new unshielded collider $C \rightarrow R \leftarrow B$ in $\mathcal{M}$ relative to $\mathbb{M}_i$ because there is $B \circ\!\!-\!\!\circ R$ in $\mathbb{M}_i$, hence $\mathcal{M}$ is not consistent to $\mathcal{P}$. Hence there does not exist an MAG consistent to $\mathbb{M}_i$ and $\mathbf{C}$. Due to the correctness of local BK, there does not exist an MAG consistent to $\mathbb{M}_i$, contradicting with Lemma 8. With (i) and (ii), it is concluded that $\mathcal{R}_2$ is not triggered in the third step of Alg. 1. Combining the parts above, we conclude that for the first edge in the third step that is transformed from $A \leftarrow\!\!\circ R$ to

$A \leftrightarrow R$ or transformed from a circle edge to a directed or bi-directed edge cannot be a circle edge, the first edge cannot be a circle edge.

Combining the two parts above, we conclude that for the first edge in the third step that is transformed from $A \leftarrow\!\circ R$ to $A \leftrightarrow R$ or transformed from a circle edge to a directed or bi-directed edge, the first edge can be neither an edge transformed from $A \leftarrow\!\circ R$ to $A \leftrightarrow R$, nor a circle edge. Hence we conclude there cannot be an edge in the third step that is transformed from $A \leftarrow\!\circ R$ to $A \leftrightarrow R$ or transformed from a circle edge to a directed or bi-directed edge. Hence, in the third step of Alg. 1, only the transformation as $A \leftarrow\!\circ R$ to $A \leftarrow R$ is possibly triggered by the orientation rules. $\qquad \square$

**Lemma 13.** *The PMG $\mathbb{M}_{i+1}$ in Thm. 1 satisfies the chordal property.*

*Proof.* We denote the oriented graph based on $\mathbb{M}_i$ and the local BK dictated by $\mathbf{C}$ after the first two steps of Alg. 1 by $\bar{\mathbb{M}}_{i+1}$. As shown by Lemma 12, there are no circle edges oriented in the third step of Alg. 1. Hence, it suffices to prove that the circle component in $\bar{\mathbb{M}}_{i+1}$ is chordal.

Note the edges in $\bar{\mathbb{M}}_{i+1}[-\mathrm{PossDe}(X, \mathbb{M}_i[-\mathbf{C}])]$ are identical to those in $\mathbb{M}_i[-\mathrm{PossDe}(X, \mathbb{M}_i[-\mathbf{C}])]$ since we do not orient the edges in this region in the first two steps. Due to chordal property of $\mathbb{M}_i$ and the fact that the subgraph of a chordal graph is also chordal, the circle component in $\bar{\mathbb{M}}_{i+1}[-\mathrm{PossDe}(X, \mathbb{M}_i[-\mathbf{C}])]$ is chordal. We consider the circle edge connecting $\bar{\mathbb{M}}_{i+1}[\mathrm{PossDe}(X, \mathbb{M}_i[-\mathbf{C}])]$ and $\bar{\mathbb{M}}_{i+1}[-\mathrm{PossDe}(X, \mathbb{M}_i[-\mathbf{C}])]$. Suppose an edge of $V_1 \circ\!\!-\!\!\circ V_2$, where $V_1 \in \mathrm{PossDe}(X, \mathbb{M}_i[-\mathbf{C}])$ and $V_2 \in \mathbf{V}(\mathbb{M}_i)\backslash\mathrm{PossDe}(X, \mathbb{M}_i[-\mathbf{C}])$. If there is $V_2 \notin \mathbf{C}$, then there is $V_2 \in \mathrm{PossDe}(X, \mathbb{M}_i[-\mathbf{C}])$ due to $V_1 \in \mathrm{PossDe}(X, \mathbb{M}_i[-\mathbf{C}])$, $V_1 \circ\!\!-\!\!\circ V_2$, and Lemma 2, contradicting with $V_2 \in \mathbf{V}(\mathbb{M}_i)\backslash\mathrm{PossDe}(X, \mathbb{M}_i[-\mathbf{C}])$. Hence there is $V_2 \in \mathbf{C}$. According to the first step of Alg. 1, $V_1 \circ\!\!-\!\!\circ V_2$ is oriented as $V_1 \leftarrow\!\circ V_2$. Hence after the first step there is not circle edge connecting $\bar{\mathbb{M}}_{i+1}[\mathrm{PossDe}(X, \mathbb{M}_i[-\mathbf{C}])]$ and $\bar{\mathbb{M}}_{i+1}[-\mathrm{PossDe}(X, \mathbb{M}_i[-\mathbf{C}])]$. There are not circle edges connecting $X$ with any other vertices in $\bar{\mathbb{M}}_{i+1}$ since the marks at $X$ is definite after the first step. In the following, it suffices to show the circle component in $\bar{\mathbb{M}}_{i+1}[\mathrm{PossDe}(X, \mathbb{M}_i[-\mathbf{C}])\backslash\{X\}]$ is chordal. For simplicity, we denote $\bar{\mathbb{M}}_{i+1}[\mathrm{PossDe}(X, \mathbb{M}_i[-\mathbf{C}])\backslash\{X\}]$ by $\bar{\mathbb{M}}_{i+1}^1$.

We will use three facts in the following: **(i)** each circle edge in $\bar{\mathbb{M}}_{i+1}$ is also a circle edge in $\mathbb{M}_i$; **(ii)** the circle edges in $\mathbb{M}_i[\mathrm{PossDe}(X, \mathbb{M}_i[-\mathbf{C}])\backslash\{X\}]$ are only possibly oriented in the second step in the process of obtaining $\bar{\mathbb{M}}_{i+1}$ from $\mathbb{M}_i$ by the first two steps of Alg. 1; **(iii)** Lemma 11.

Suppose the circle component in $\bar{\mathbb{M}}_{i+1}^1$ is not chordal, there is a circle cycle as $V_0 \circ\!\!-\!\!\circ V_1 \circ\!\!-\!\!\circ \cdots \circ\!\!-\!\!\circ V_n \circ\!\!-\!\!\circ V_0, n \geq 3$, where there is not a circle edge between every two unconsecutive vertices. And there must exist edges between the unconsecutive vertices in this cycle, otherwise it is a cycle of length four or more without a chord in $\mathbb{M}_i$, contradicting with the chordal property of $\mathbb{M}_i$. We can always find a sub-structure $V_k \circ\!\!-\!\!\circ V_{k+1} \circ\!\!-\!\!\circ \cdots \circ\!\!-\!\!\circ V_m \leftarrow V_k, 0 \leq k < m \leq n$ without other directed edges between any two vertices among $V_k, \cdots, V_m$ except for $V_m \leftarrow V_k$ (if there is another directed edge, for instance $V_{k+1} \to V_m$, we can find a proper sub-structure $V_{k+1} \circ\!\!-\!\!\circ \cdots \circ\!\!-\!\!\circ V_m \leftarrow V_{k+1}$ instead. And since the path is symmetric, suppose $V_k \to V_m$ without loss of generality.). According to Lemma 3, the directed edge between $V_k$ and $V_m$ can only be a circle edge in $\mathbb{M}_i$. Hence in $\mathbb{M}_i$ there is $V_k \circ\!\!-\!\!\circ V_{k+1} \circ\!\!-\!\!\circ \cdots \circ\!\!-\!\!\circ V_m \circ\!\!-\!\!\circ V_k$. Since the circle component in $\mathbb{M}_i$ in chordal, the length of the circle cycle can only be three. Hence it holds $m = k + 2$ and there is a sub-structure $V_k \circ\!\!-\!\!\circ V_{k+1} \circ\!\!-\!\!\circ V_{k+2} \leftarrow V_k$ in $\bar{\mathbb{M}}_{i+1}^1$. Next, we will prove its impossibility.

Since there is $V_k \circ\!\!-\!\!\circ V_{k+1} \circ\!\!-\!\!\circ V_{k+2} \leftarrow V_k$ in $\bar{\mathbb{M}}_{i+1}^1$, there is $\mathcal{F}_{V_k} = \mathcal{F}_{V_{k+1}} = \mathcal{F}_{V_{k+2}}$. Considering it is oriented as $V_k \to V_{k+2}$ in the second step, there is another vertex $F_1$ in $\bar{\mathbb{M}}_{i+1}^1$ such that there is $F_1 \to V_k$ where $F_1$ is not adjacent to $V_{k+2}$. Evidently $F_1$ is adjacent to $V_{k+1}$, otherwise $V_k \to V_{k+1}$ is also oriented. Next, we consider the relation between $\mathcal{F}_{F_1}$ and $\mathcal{F}_{V_k}$. Since $F_1 \to V_k$, there is $\mathcal{F}_{V_k} \subseteq \mathcal{F}_{F_1}$ (Note $F_1 \to V_k$ is oriented in the second step, if $\mathcal{F}_{V_k} \supset \mathcal{F}_{F_1}$, there can be an another edge oriented as $V_k \to F_1$ in the second step, contradicting with Lemma 11). If $\mathcal{F}_{V_k} \subset \mathcal{F}_{F_1}$, there is also $\mathcal{F}_{V_{k+1}} \subset \mathcal{F}_{F_1}$ since $\mathcal{F}_{V_k} = \mathcal{F}_{V_{k+1}}$. Hence $F_1 \to V_{k+1}$. And due to $\mathcal{F}_{V_{k+1}} = \mathcal{F}_{V_{k+2}}$ and the non-adjacency of $F_1$ and $V_{k+2}$, in the second step $V_{k+1} \to V_{k+2}$ is oriented, contradicting with $V_{k+1} \circ\!\!-\!\!\circ V_{k+2}$. Hence, it is only possible that there is $\mathcal{F}_{V_k} = \mathcal{F}_{F_1}$ and $F_1 \circ\!\!-\!\!\circ V_{k+1}$ in $\bar{\mathbb{M}}_{i+1}^1$. Here we find a sub-structure $F_1 \circ\!\!-\!\!\circ V_{k+1} \circ\!\!-\!\!\circ V_k \leftarrow F_1$. Since $F_1 \to V_k$ is oriented, there is another vertex $F_2$ that is not adjacent to $V_k$ in $\bar{\mathbb{M}}_{i+1}^1$ such that $F_2 \to F_1$ is oriented in the second step. Similar to the previous proof, there is not a contradiction only when $\mathcal{F}_{F_2} = \mathcal{F}_{F_1}$ and $F_2 \circ\!\!-\!\!\circ V_{k+1}$. Repeat this process and we can conclude that if there is not a contradiction, in any uncovered directed path as $F_t \to \cdots \to F_1 \to V_k \to V_{k+2}$, for any a vertex $V'$ on the path, there is $\mathcal{F}_{V'} = \mathcal{F}_{V_k}$ and

there is a circle edge between $V'$ and $V_{k+1}$. It contradicts with Lemma 7. Hence, there cannot be a sub-structure as $V_k \circ\!\!-\!\!\circ V_{k+1} \circ\!\!-\!\!\circ V_{k+2} \leftarrow V_k$ after the first three steps. The proof completes. $\qquad\square$

**Lemma 14.** *The PMG $\mathbb{M}_{i+1}$ in Thm. 1 satisfies the balanced property.*

*Proof.* If there is $V_i *\!\!\rightarrow V_j \circ\!\!-\!\!* V_k$, we first prove that $V_i$ is adjacent to $V_k$. Suppose $V_i$ is not adjacent to $V_k$. This structure cannot appear in $\mathbb{M}_i$ due to the complete property of $\mathbb{M}_i$. Hence $V_i *\!\!\rightarrow V_j$ is oriented in the process of obtaining $\mathbb{M}_{i+1}$ based on $\mathbb{M}_i$. According to Lemma 12, the arrowhead is introduced in only the first two steps of obtaining $\mathbb{M}_{i+1}$. And in the first two steps arrowhead is added at the vertex in $\mathrm{PossDe}(X, \mathbb{M}_i[-\mathbf{C}])$. Hence there is $V_j \in \mathrm{PossDe}(X, \mathbb{M}_i[-\mathbf{C}])$. In this case if $V_k \in \mathbf{C}$, there is $V_j \leftarrow\!\!* V_k$ oriented in the first step, contradiction. If $V_k \notin \mathbf{C}$, if there is $V_j \circ\!\!-\!\!\circ V_k$ in $\mathbb{M}_i$, there is $V_k \in \mathrm{PossDe}(X, \mathbb{M}_i[-\mathbf{C}])$, thus there is always $V_j \rightarrow V_k$ oriented in the second step by discussing whether $V_i \in \mathbf{C}$ (we omit the details), contradiction. If $V_j \circ\!\!\rightarrow V_k$ in $\mathbb{M}_i$, it will be oriented as $V_j \rightarrow V_k$ by $\mathcal{R}_1$ in the third step of Alg. 1, which contradict with $V_j \circ\!\!-\!\!* V_k$ in $\mathbb{M}_{i+1}$.

Next we consider the case that $V_i$ is adjacent to $V_k$. If there is $V_i *\!\!\rightarrow V_j \circ\!\!-\!\!* V_k$ in $\mathbb{M}_i$, there is $V_i *\!\!\rightarrow V_k$ due to the balanced property of $\mathbb{M}_i$, hence there is $V_i *\!\!\rightarrow V_k$ in $\mathbb{M}_{i+1}$. Hence it suffices to consider there is $V_i *\!\!-\!\!\circ V_j \circ\!\!-\!\!* V_k$ in $\mathbb{M}_i$ while $V_i *\!\!\rightarrow V_j \circ\!\!-\!\!* V_k$ in $\mathbb{M}_{i+1}$. Note in the process of obtaining $\mathbb{M}_{i+1}$ based on $\mathbb{M}_i$, arrowhead are oriented only at the vertex in $\mathrm{PossDe}(X, \mathbb{M}_i[-\mathbf{C}])$, thereby $V_j \in \mathrm{PossDe}(X, \mathbb{M}_i[-\mathbf{C}])$. In addition, $V_k \notin \mathbf{C}$, otherwise there is $V_j \leftarrow\!\!* V_k$ in $\mathbb{M}_{i+1}$. Combining $V_j \circ\!\!-\!\!* V_k$ and $V_j \in \mathrm{PossDe}(X, \mathbb{M}_i[-\mathbf{C}])$, there is $V_k \in \mathrm{PossDe}(X, \mathbb{M}_i[-\mathbf{C}])$. We discuss whether $V_i \in \mathbf{C}$ in the following.

**(i).** If $V_i \in \mathbf{C}$, there is $V_i *\!\!\rightarrow V_j$ and $V_i *\!\!\rightarrow V_k$ after the first step of obtaining $\mathbb{M}_{i+1}$ based on $\mathbb{M}_i$. In this case, when there is $V_i \rightarrow V_j$ in $\mathbb{M}_{i+1}$, there is $V_i \circ\!\!-\!\!\circ V_j$ in $\mathbb{M}_i$, hence there is $V_i \circ\!\!-\!\!* V_k$ or $V_i \rightarrow V_k$ in $\mathbb{M}_i$ (if there is $V_i \leftarrow\!\!* V_k \circ\!\!-\!\!* V_j \circ\!\!-\!\!* V_i$ in $\mathbb{M}_i$, it contradicts with the balanced property of $\mathbb{M}_i$). Therefore there is $V_i \rightarrow V_k$ in $\mathbb{M}_{i+1}$. Balanced property is satisfied for $\mathbb{M}_{i+1}$ when $V_i \in \mathbf{C}$.

**(ii).** If $V_i \notin \mathbf{C}$, there is $V_i \in \mathrm{PossDe}(X, \mathbb{M}_i[-\mathbf{C}])$ due to $V_i *\!\!-\!\!\circ V_j$ and $V_j \in \mathrm{PossDe}(X, \mathbb{M}_i[-\mathbf{C}])$. In this case the arrowhead is introduced in the second step of obtaining $\mathbb{M}_{i+1}$ based on $\mathbb{M}_i$. Hence there is $V_i \circ\!\!-\!\!\circ V_j$ in $\mathbb{M}_i$. In this case either $V_i \circ\!\!-\!\!\circ V_j *\!\!\rightarrow V_k \leftarrow\!\!* V_i$, or $V_i \circ\!\!-\!\!\circ V_j \circ\!\!-\!\!\circ V_k \circ\!\!-\!\!\circ V_i$ in $\mathbb{M}_i$. For the former case, there is $V_i \rightarrow V_j *\!\!\rightarrow V_k \leftarrow\!\!* V_i$ in $\mathbb{M}_{i+1}$. And there cannot be $V_i \leftrightarrow V_k$ in $\mathbb{M}_{i+1}$, otherwise there is $V_j \leftrightarrow V_k$ since $\mathbb{M}_{i+1}$ is closed under $\mathcal{R}_2$, contradicting with $V_j \circ\!\!-\!\!* V_k$ in $\mathbb{M}_{i+1}$. Hence the balanced property also holds in $\mathbb{M}_{i+1}$ for the first case. For the latter case, suppose there is $V_i \rightarrow V_j \circ\!\!-\!\!\circ V_k$ oriented after the second step, $V_i$ is adjacent to $V_k$. According to the proof of Lemma 13, there cannot be a structure $V_i \circ\!\!-\!\!\circ V_k \circ\!\!-\!\!\circ V_j \leftarrow V_i$, thus there is not a circle-edge between $V_i$ and $V_k$. Since we only transform the circle edges between $\mathrm{PossDe}(X, \mathbb{M}_i[-\mathbf{C}])$ to directed edges in the second step, the edge between $V_i$ and $V_k$ is directed. If there is $V_i \rightarrow V_k$ oriented in the second step, balanced property of $\mathbb{M}_{i+1}$ is satisfied. If there is $V_k \rightarrow V_i$ oriented in the second step, there is $V_k \rightarrow V_j$ oriented by $\mathcal{R}_2$ in the third step, which contradicts with Lemma 12 that there are no circle edges oriented in the third step, impossibility.

As shown above, balanced property also holds in $\mathbb{M}_{i+1}$. $\qquad\square$

**Lemma 15.** *The PMG $\mathbb{M}_{i+1}$ in Thm. 1 satisfies the complete property.*

In $\mathbb{M}_{i+1}$, the edges with circles are either $A \circ\!\!-\!\!\circ B$ or $A \circ\!\!\rightarrow B$. In Lemma 15.1, we show that we can always obtain an MAG consistent to $\mathcal{P}$ and local BK regarding $V_1, \cdots, V_{i+1}$ by transforming $\circ\!\!\rightarrow$ to $\rightarrow$ and the circle component into a DAG without unshielded colliders in $\mathbb{M}_{i+1}$. Due to the chordal property of $\mathbb{M}_{i+1}$, for the edge $A \circ\!\!-\!\!\circ B$ in $\mathbb{M}_{i+1}$, there exist both perfect elimination orders to orient the circle component into DAGs without unshielded colliders where there is $A \rightarrow B$ and $A \leftarrow B$ respectively, as implied by Lemma 5 of Meek [36]; and for the edge $C \circ\!\!\rightarrow D$ in $\mathbb{M}_{i+1}$, the edge can be oriented as $C \rightarrow D$ in some MAG consistent to $\mathcal{P}$ and local BK regarding $V_1, \cdots, V_{i+1}$. For the edge $A \circ\!\!\rightarrow B$ in $\mathbb{M}_{i+1}$, we show the edge can be oriented as $A \leftrightarrow B$ in Lemma 15.2. Here the most difficult part is to prove Lemma 15.1, *i.e.*, we can always obtain an MAG consistent to $\mathcal{P}$ and local BK regarding $V_1, \cdots, V_{i+1}$ by transforming $\circ\!\!\rightarrow$ to $\rightarrow$ and the circle component into a DAG without unshielded colliders in $\mathbb{M}_{i+1}$. With this result, we can prove Lemma 15.2 totally following the procedure of that of Theorem 3 of Zhang [35], with the invariant, chordal, and balanced property of $\mathbb{M}_{i+1}$. Since the proof is too lengthy and completely follow that of Theorem 3 of Zhang [35], we will not show the details but just a sketch in the proof of Lemma 15.2.

Note according to the invariant property of $\mathbb{M}_{i+1}$, there cannot be new unshielded colliders or directed or almost directed cycles introduced in $\mathbb{M}_{i+1}$ relative to $\mathbb{M}_i$, otherwise there does not exist an MAG

consistent to $\mathbb{M}_{i+1}$. Given the invariant property of $\mathbb{M}_{i+1}$ by Lemma 6 and the basic assumption that the background knowledge is correct, there is not an MAG consistent to $\mathbb{M}_i$, contradicting with Lemma 8.

**Lemma 15.1.** *Consider $\mathbb{M}_{i+1}$ in Thm. 1. We orient a graph $\mathcal{H}$ from $\mathbb{M}_{i+1}$ by transforming $\circ\!\to$ to $\to$ and the circle component in $\mathbb{M}_{i+1}$ into a DAG without unshielded colliders. Then $\mathcal{H}$ is an MAG consistent to $\mathcal{P}$ and local BK regarding $V_1, \cdots, V_{i+1}$.*

*Proof.* Our proof idea is as follows. We aim to prove if the constructed graph $\mathcal{H}$ by the following orientation process based on $\mathbb{M}_{i+1}$ which transforms all edges $\circ\!\to$ to $\to$ and orient the circle component into a DAG without unshielded colliders is not an MAG consistent to $\mathcal{P}$ and local BK regarding $V_1, \cdots, V_{i+1}$, then a constructed graph based on $\mathbb{M}_i$ which transform all edges $\circ\!\to$ to $\to$ and orient the circle component into a DAG without unshielded colliders is not an MAG consistent to $\mathcal{P}$ and local BK regarding $V_1, \cdots, V_i$. Repeat this process until $\mathbb{M}_0$ and we can conclude that a constructed graph based on $\mathbb{M}_0(=\mathcal{P})$ which transforms all edges $\circ\!\to$ to $\to$ and orient the circle component into a DAG without unshielded colliders is not an MAG consistent to $\mathbb{M}_0(=\mathcal{P})$, which contradicts Theorem 2 of Zhang [35]. Hence we get the desired result. The process of obtaining $\mathcal{H}$ is as follows.

(Step 1) For all $K \in \mathrm{PossDe}(X, \mathbb{M}_i[-\mathbf{C}])$ and $\forall T \in \mathbf{C}$ such that $K \circ\!\!-\!\ast T$ in $\mathbb{M}_i$, orient $K \leftarrow\!\ast T$ (the mark at $T$ remains); for all $K \in \mathrm{PossDe}(X, \mathbb{M}_i[-\mathbf{C}])$ such that $X \circ\!\!-\!\ast K$, orient $X \to K$;

(Step 2) Orient the subgraph $\mathbb{M}_i[\mathrm{PossDe}(X, \mathbb{M}_i[-\mathbf{C}])\backslash\{X\}]$ as follows until no feasible updates: for any two vertices $V_i$ and $V_j$ such that $V_i \circ\!\!-\!\!\circ V_j$, orient it as $V_i \to V_j$ if **(i)** $\mathcal{F}_{V_i}\backslash\mathcal{F}_{V_j} \neq \emptyset$ or **(ii)** $\mathcal{F}_{V_i} = \mathcal{F}_{V_j}$ as well as there is a vertex $V_k \in \mathrm{PossDe}(X, \mathbb{M}_i[-\mathbf{C}])\backslash\{X\}$ not adjacent to $V_j$ such that $V_k \to V_i \circ\!\!-\!\!\circ V_j$, where $\mathcal{F}_{V_l} = \{V \in \mathbf{C} \cup \{X\} \mid V \ast\!\!-\!\!\circ V_l$ in $\mathbb{M}_i\}$;

(Step 3) Obtain $\mathbb{M}_{i+1}$ by applying the orientation rules until the graph is closed under the rules;

(Step 4) for the circle component in subgraph $\mathbb{M}_{i+1}[\mathrm{PossDe}(X, \mathbb{M}_i[-\mathbf{C}])\backslash\{X\}]$, orient it into a DAG without new unshielded colliders;

(Step 5) for the circle component in subgraph $\mathbb{M}_{i+1}[-\mathrm{PossDe}(X, \mathbb{M}_i[-\mathbf{C}])]$, orient it into a DAG without new unshielded colliders;

(Step 6) for any edge as $\circ\!\to$, orient it as $\to$.

Note the first three steps are the process of obtaining $\mathbb{M}_{i+1}$ from $\mathbb{M}_i$ with the local structure of $X$ dictated by $\mathbf{C}$. And in Step 4 - Step 6 we transform the edges as $\circ\!\to$ to $\to$ and transform the circle component in $\mathbb{M}_{i+1}$ into a DAG without new unshielded colliders. Note $X$ is not in a connected circle component with any other vertices after the first step. Hence when we consider the circle component in $\mathbb{M}_{i+1}$, we do not need to consider $X$. And we have proven that there is not a circle edge connecting $\mathrm{PossDe}(X, \mathbb{M}_i[-\mathbf{C}])\backslash\{X\}$ and $\mathbf{V}(\mathbb{M}_i)\backslash(\mathrm{PossDe}(X, \mathbb{M}_i[-\mathbf{C}])\backslash\{X\})$ in the proof of Lemma 13. Hence the circle component in $\mathbb{M}_{i+1}[\mathrm{PossDe}(X, \mathbb{M}_i[-\mathbf{C}])\backslash\{X\}]$ is not connected to that in $\mathbb{M}_{i+1}[-(\mathrm{PossDe}(X, \mathbb{M}_i[-\mathbf{C}])\backslash\{X\})]$. Hence we can divide the circle component orientation in $\mathbb{M}_{i+1}$ into Step 4 and Step 5. The achievability of Step 4 and Step 5 is due to the chordal property of $\mathbb{M}_{i+1}$ according to Lemma 13.

In the following there are mainly two parts. The first part is that we construct an auxiliary graph $\mathcal{H}_0$ *based on* $\mathcal{H}$, and we show that this constructed graph can also be seen as a graph obtained from $\mathbb{M}_i$ by transforming all edges $\circ\!\to$ to $\to$ and orienting the circle component into a DAG without new unshielded colliders. The second part is we show that if $\mathcal{H}_0$ is an MAG consistent to $\mathcal{P}$ and local BK regarding $V_1, \cdots, V_i$, then $\mathcal{H}$ is an MAG consistent to $\mathcal{P}$ and local BK regarding $V_1, \cdots, V_{i+1}$.

**(A) Auxiliary graph $\mathcal{H}_0$.** We construct an auxiliary graph $\mathcal{H}_0$ based on $\mathcal{H}$ by transforming all and only the bi-directed edges $K \leftrightarrow T$ in $\mathbb{M}_{i+1}$ which are $K \circ\!\to T$ in $\mathbb{M}_i$ to $K \to T$, where $K \in \mathrm{PossDe}(X, \mathbb{M}_i[-\mathbf{C}])$ and $T \in \mathbf{C}$ according to the first step. It is direct that $\mathcal{H}_0$ has the non-circle marks in $\mathbb{M}_i$ and there are no new bi-directed edges in $\mathcal{H}_0$ compared to $\mathbb{M}_i$ because all additional bi-directed edges in $\mathcal{H}$ relative to $\mathbb{M}_i$ are possibly introduced in only the first step of the process of obtaining $\mathcal{H}$ according to the construction process and Lemma 12. Besides, all the circles on $\circ\!\to$ edges in $\mathbb{M}_i$ are oriented as tails in $\mathcal{H}_0$. In the following it suffices to show that $\mathcal{H}_0$ is also a

graph oriented from $\mathbb{M}_i$ by orienting the circle component in $\mathbb{M}_i$ into a DAG without new unshielded colliders.

Hence, we only consider the circle component in $\mathbb{M}_i$. We divide it into two parts, one is the circle component in $\mathbb{M}_i[\text{PossDe}(X, \mathbb{M}_i[-\mathbf{C}])\backslash\{X\}]$, denoted by $\text{CC}_1$; and the other is the circle component in $\mathbb{M}_i[-(\text{PossDe}(X, \mathbb{M}_i[-\mathbf{C}])\backslash\{X\})]$, denoted by $\text{CC}_2$.

Note the oriented edges of $\text{CC}_1$ in $\mathcal{H}_0$ totally follows those in $\mathcal{H}$, which are oriented by either Step 2 or Step 4. There are no new unshielded colliders or directed or almost directed cycles oriented in the edges of $\text{CC}_1$ by the three following facts. **(1).** There are no new unshielded colliders or directed or almost directed cycles in the edges of $\text{CC}_1$ oriented by Step 2. Otherwise, given the invariant property of $\mathbb{M}_{i+1}$ by Lemma 6 and the basic assumption that the background knowledge is correct, there is not an MAG consistent to $\mathbb{M}_i$, contradicting with Lemma 8. **(2).** There are no unshielded colliders or directed or almost directed cycles in the edges of $\text{CC}_1$ oriented by Step 4 because the circle component in $\mathbb{M}_{i+1}$ is chordal and is oriented to a DAG without new unshielded colliders. **(3).** There are no new unshielded colliders or directed or almost directed cycles in edges of $\text{CC}_1$ oriented by both Step 2 and Step 4 due to the balanced property of $\mathbb{M}_{i+1}$ and the impossibility of the transformation of circle edges to bi-directed edges.

Note the edges in $\text{CC}_2$ also totally follow those in $\mathcal{H}$. Although when $X \circ\!\!\rightarrow T$ in $\mathbb{M}_i$ where $T \in \mathbf{C}$, there is $X \leftrightarrow T$ in $\mathcal{H}$ while $X \rightarrow T$, such edge is not in the circle component $\text{CC}_2$ because it is as $X \circ\!\!\rightarrow T$ in $\mathbb{M}_i$. According to the orientation process, the sub circle component of $\text{CC}_2$ induced by $\mathbf{V}(\text{CC}_2)\backslash\{X\}$, is oriented into a DAG without new unshielded colliders. Hence if there are new unshielded colliders or directed or almost directed cycles in edges of $\text{CC}_2$, they contain $X$. **(1)** There are not new unshielded colliders as $A*\!\!\rightarrow X \leftarrow\!\!*B$ in edges of $\text{CC}_2$ in Step 2. Otherwise, given the invariant property of $\mathbb{M}_{i+1}$ by Lemma 6 and the basic assumption that the background knowledge is correct, there is not an MAG consistent to $\mathbb{M}_i$, contradicting with Lemma 8. **(2)** There are no directed or almost directed cycles in $\text{CC}_2$ containing $X$ because for each vertex $V$ in $\text{CC}_2$ that has a circle edge with $X$, the edge is oriented as $V \rightarrow X$.

For the circle edge in the circle component connecting $K \in \text{PossDe}(X, \mathbb{M}_i[-\mathbf{C}])\backslash\{X\}$ and $T \in \mathbf{V}(\mathbb{M}_i)\backslash(\text{PossDe}(X, \mathbb{M}_i[-\mathbf{C}])\backslash\{X\})$, there must be $T \in \mathbf{C} \cup \{X\}$, otherwise there is $T \in \text{PossDe}(X, \mathbb{M}_i[-\mathbf{C}])\backslash\{X\}$ due to $K \in \text{PossDe}(X, \mathbb{M}_i[-\mathbf{C}])\backslash\{X\}$ and $K \circ\!\!-\!\!\circ T$. Hence in $\mathcal{H}$ the circle edge is oriented as $K \leftarrow T$ by the first step and the last step. According to the relation between $\mathcal{H}$ and $\mathcal{H}_0$, there is $K \leftarrow T$ in $\mathcal{H}_0$. Hence, for each circle edge $K \circ\!\!-\!\!\circ T$ where $K \in \text{PossDe}(X, \mathbb{M}_i[-\mathbf{C}])\backslash\{X\}$ and $T \in \mathbf{V}(\mathbb{M}_i)\backslash(\text{PossDe}(X, \mathbb{M}_i[-\mathbf{C}])\backslash\{X\})$, there is $K \leftarrow T$ in $\mathcal{H}_0$ and $T \in \mathbf{C} \cup \{X\}$. Hence it is evident that in $\mathcal{H}_0$ there cannot be a directed or almost directed cycles oriented from the circle component which contain both the vertices in $\text{PossDe}(X, \mathbb{M}_i[-\mathbf{C}])\backslash\{X\}$ and $\mathbf{V}(\mathbb{M}_i)\backslash(\text{PossDe}(X, \mathbb{M}_i[-\mathbf{C}])\backslash\{X\})$. If there is a new unshielded collider in $\mathcal{H}_0$ relative to $\mathbb{M}_i$ comprised of the vertices in both $\text{PossDe}(X, \mathbb{M}_i[-\mathbf{C}])\backslash\{X\}$ and $\mathbf{V}\backslash(\text{PossDe}(X, \mathbb{M}_i[-\mathbf{C}])\backslash\{X\})$, the unshielded collider can only be as $T_1 \rightarrow K_1 \leftarrow T_2$ or $T_1 \rightarrow K_1 \leftarrow K_2$ where $T_1, T_2 \in \mathbf{C} \cup \{X\}$ and $K_1, K_2 \in \text{PossDe}(X, \mathbb{M}_i[-\mathbf{C}])\backslash\{X\}$. We first prove that the first case is impossible. Due to Lemma 6, there is $T_1*\!\!\rightarrow K_1$ and $T_2*\!\!\rightarrow K_2$ in all MAGs consistent to $\mathcal{P}$ and the local BK regarding $V_1, \ldots, V_{i+1}$. If they form a new unshielded collider, it implies that there does not exist MAG consistent to $\mathcal{P}$ and the local BK regarding $V_1, \ldots, V_{i+1}$. Due to the correctness of BK and Lemma 6, there does not exist an MAG consistent to $\mathbb{M}_i$, contradicting with Lemma 8. For the second case, there is $T_1 \in \mathcal{F}_{K_1}\backslash\mathcal{F}_{K_2}$, hence $K_1 \rightarrow K_2$ should be oriented in the second step. While there is also $K_1 \leftarrow K_2$ oriented, it contradicts with Lemma 11. Hence, if there is a new unshielded collider in $\mathcal{H}_0$, there is always a contradiction.

Hence, we prove that the graph $\mathcal{H}_0$ constructed based on $\mathcal{H}$ can also be seen as a graph obtained from $\mathbb{M}_i$ by transforming all edges $\circ\!\!\rightarrow$ to $\rightarrow$ and transforming the circle component into a DAG without new unshielded colliders.

**(B) If $\mathcal{H}_0$ is an MAG consistent to $\mathcal{P}$ and local BK regarding $V_1, \cdots, V_i$, then $\mathcal{H}$ is an MAG consistent to $\mathcal{P}$ and local BK regarding $V_1, \cdots, V_{i+1}$.** Suppose $\mathcal{H}_0$ is an MAG consistent to $\mathcal{P}$ and local BK regarding $V_1, \cdots, V_i$. We will prove that $\mathcal{H}$ is an MAG Markov equivalent to $\mathcal{H}_0$ by Lemma 1 of Zhang and Spirtes [41]. Because $\mathcal{H}$ has the non-circle marks in $\mathbb{M}_{i+1}$, and $\mathcal{H}_0$ belongs to the MEC represented by $\mathcal{P}$, we can conclude that $\mathcal{H}$ is an MAG consistent to $\mathcal{P}$ and local BK regarding $V_1, \cdots, V_{i+1}$.

Note that the only difference between $\mathcal{H}$ and $\mathcal{H}_0$ is that for $\forall K \in \text{PossDe}(X, \mathbb{M}_i[-\mathbf{C}])$ and $\forall T \in \mathbf{C}$ such that $K \circ\!\!\rightarrow T$ in $\mathbb{M}_i$, there is $K \rightarrow T$ in $\mathcal{H}_0$ but $K \leftrightarrow T$ in $\mathcal{H}$. Denote the set of different edges in $\mathcal{H}_0$ by $Edge(\mathcal{H}_0) = \{K \rightarrow T \text{ in } \mathcal{H}_0 \mid K \in \text{PossDe}(X, \mathbb{M}_i[-\mathbf{C}]), T \in \mathbf{C}, K \circ\!\!\rightarrow T \text{ in } \mathbb{M}_i\}$. We could obtain $\mathcal{H}$ from $\mathcal{H}_0$ by transforming these edges to bi-directed edges. We transform one edge one time. At first, we select the edge $K \rightarrow T$ in $Edge(\mathcal{H}_0)$ according to the selection criterion that (1) we select $K$ that is not an ancestor of any other $V_1$ such that there is an edge $V_1 \rightarrow V_2$ in $Edge(\mathcal{H}_0)$; and (2) given $K$ selected in the first step, we select $T$ that is not a descendant of any other $V_2$ such that there is an edge $K \rightarrow V_2$ in $Edge(\mathcal{H}_0)$. Then we obtain $Edge(\mathcal{H}_1)$ by deleting $K \rightarrow T$ from $Edge(\mathcal{H}_0)$. By such operation, we obtain a new graph $\mathcal{H}_1$ and $Edge(\mathcal{H}_1)$. Repeat the process above and we could obtain a series of graphs $\mathcal{H}_0, \mathcal{H}_1, \cdots, \mathcal{H}_m, \mathcal{H}_{m+1}(= \mathcal{H})$. We will prove that for any $\mathcal{H}_j$ and $\mathcal{H}_{j+1}$, where $0 \leq j \leq m$, if $\mathcal{H}_j$ is an MAG, then $\mathcal{H}_{j+1}$ is an MAG Markov equivalent to $\mathcal{H}_j$. According to the conditions, $\mathcal{H}_0$ is an MAG in the MEC represented by $\mathcal{P}$. Suppose the edge that will be transformed in $\mathcal{H}_j$ is $K \rightarrow T$. According to Lemma 1 of Zhang and Spirtes [41], given $\mathcal{H}_j$ is an MAG, it suffices to show that (1) there is no directed path from $K$ to $T$ in $\mathcal{H}_j$ other than $K \rightarrow T$; (2) for any $A \rightarrow K$ in $\mathcal{H}_j$, $A \rightarrow T$ is also in $\mathcal{H}_j$; and for any $B \leftrightarrow K$ in $\mathcal{H}_j$, either $B \rightarrow T$ or $B \leftrightarrow T$ is in $\mathcal{H}_j$; (3) there is no discriminating path for $K$ on which $T$ is the endpoint adjacent to $K$ in $\mathcal{H}_j$. We show the proof in order.

(1) In this part, we prove that there is not a directed path from $K$ to $T$ in $\mathcal{H}_j$. For the sake of contradiction, suppose there is a directed path from $K$ to $T$ in $\mathcal{H}_j$, we suppose the minimal directed path of this path is $K(= F_0) \rightarrow F_1 \rightarrow \cdots \rightarrow F_m \rightarrow T(= F_{m+1})$. Since we only transform directed edges to bi-directed edges in the process, the directed path is also in $\mathcal{H}_0$. We first prove that there must be a vertex $F_n, 1 \leq n \leq m$ such that $F_n \in \mathbf{C}$. Otherwise, all of $F_1, \cdots, F_m$ belong to $\text{PossDe}(X, \mathbb{M}_i[-\mathbf{C}])$ since $F_0 \in \text{PossDe}(X, \mathbb{M}_i[-\mathbf{C}])$ and there is a possible directed path comprised of $F_0, F_1, \cdots, F_m$ in $\mathbb{M}_i$. **(i.)** If there is $F_m \rightarrow T$ in $\mathbb{M}_i$, it contradicts with Lemma 10. **(ii.)** If there is $F_m \circ\!\!-\!\!\circ T$ in $\mathbb{M}_i$, according to the first step of orientation procedure to construct $\mathcal{H}$, there is $F_m \leftarrow T$ in $\mathcal{H}$. Since in the process from $\mathcal{H}_j$ to $\mathcal{H}$ we never transform an edge $A \rightarrow B$ to $A \leftarrow B$, there cannot be an edge $F_m \rightarrow T$ in $\mathcal{H}_j$. **(iii.)** If there is $F_m \circ\!\!\rightarrow T$ in $\mathbb{M}_i$, there is $F_m \rightarrow T$ in $\mathcal{H}_0$. According to the edge selection criterion, when there is both $F_m \rightarrow T$ and $K \rightarrow T$ in $\mathcal{H}_j$, we should transform $F_m \rightarrow T$ ahead of $K \rightarrow T$ due to $K \rightarrow F_1 \rightarrow \cdots \rightarrow F_m$, contradiction. For the other situations for the edge between $F_m$ and $T$ in $\mathbb{M}_i$, there cannot form an edge $F_m \rightarrow T$ in $\mathcal{H}_j$. Hence we conclude there is a vertex $F_n, 1 \leq n \leq m$ such that $F_n \in \mathbf{C}$.

Without loss of generality, we suppose $F_n \in \mathbf{C}$ and $F_l \notin \mathbf{C}, \forall 1 \leq l \leq n - 1$. We first prove there is not a vertex $F_l, 1 \leq l \leq n - 1$ adjacent to $T$. If there is, since $F_l \rightarrow \cdots \rightarrow F_m \rightarrow T$ in $\mathcal{H}_0$, there is $F_l \rightarrow T$ in $\mathcal{H}_0$ due to the ancestral property. In this case there is a directed path $F_1 \rightarrow \cdots F_l \rightarrow T$ without vertices in $\mathbf{C}$ in $\mathcal{H}_0$, which implies that there is a possible directed path where the sub-path from $F_1$ to $F_l$ is minimal and any variables on the path do not belong to $\mathbf{C}$, contradicting the result we prove above. Hence $F_l$ cannot be adjacent to $T$ for $\forall 1 \leq l \leq n - 1$. **(i.)** If $n \geq 2$, **(i.a.)** if there $F_n \circ\!\!-\!\!* T$ or $F_n \rightarrow T$ in $\mathbb{M}_i$, there is an uncovered possible directed path comprised of $K, F_1, \cdots, F_n, T$ in $\mathbb{M}_i$ where $F_1$ is not adjacent to $T$. In this case $K \circ\!\!\rightarrow T$ has been oriented as $K \rightarrow T$ in $\mathbb{M}_i$ by $\mathcal{R}_9$ of Zhang [35] due to $\mathbb{M}_i$ is closed under the orientation rules, contradiction. **(i.b.)** If there is $F_n \leftarrow\!\!* T$ in $\mathbb{M}_i$, note the non-adjacency of $T$ and $F_{n-1}$. Due to the edge $T*\!\!\rightarrow F_n$ and the complete property of $\mathbb{M}_i$, the mark at $F_n$ on the edge between $F_{n-1}$ and $F_n$ is identifiable in $\mathbb{M}_i$. And due to the possible directed path, there is $F_{n-1} \rightarrow F_n$ in $\mathcal{H}_0$, there can only be $F_{n-1} \rightarrow F_n$ or $F_{n-1} \circ\!\!\rightarrow F_n$ in $\mathbb{M}_i$. The former case contradicts with Lemma 10 due to $F_{n-1} \in \text{PossDe}(X, \mathbb{M}_i[-\mathbf{C}])$ and $F_n \in \mathbf{C}$. For the latter case, the edge $F_{n-1} \rightarrow F_n$ should be transformed to bi-directed edge ahead of $K \rightarrow T$, hence there cannot be an edge $F_{n-1} \rightarrow F_n$ in $\mathcal{H}_j$, contradiction. **(ii.)** If $n = 1$, there is $K \rightarrow T' \rightarrow T$ in $\mathcal{H}$, where $T' \in \mathbf{C}$. In this case if there is not $K \circ\!\!\rightarrow T'$ in $\mathbb{M}_i$, there cannot be an edge $K \rightarrow T'$ in $\mathcal{H}_j$; if there is $K \circ\!\!\rightarrow T'$ in $\mathbb{M}_i$, there is thus both $K \rightarrow T'$ and $K \rightarrow T$ in $\mathcal{H}_0$, $K \circ\!\!\rightarrow T'$ is transformed to a bi-directed edge ahead of $K \rightarrow T$ due to $T' \rightarrow T$, thereby there is not an edge $K \rightarrow T'$ in $\mathcal{H}_j$. Hence there cannot be a sub-structure $K \rightarrow T' \rightarrow T$ in $\mathcal{H}_j$, contradiction. Hence, there is always a contradiction if there is a directed path from $K$ to $T$ in $\mathcal{H}_j$.

(2) In this part, we prove that if there is an edge $A \rightarrow K$ in $\mathcal{H}_j$, there is $A \rightarrow T$ in $\mathcal{H}_j$; if there is $B \leftrightarrow K$ in $\mathcal{H}_j$, either $B \rightarrow T$ or $B \leftrightarrow T$ is in $\mathcal{H}_j$. Note there is $K \circ\!\!\rightarrow T$ in $\mathbb{M}_i$, where $K \in \text{PossDe}(X, \mathbb{M}_i[-\mathbf{C}])$ and $T \in \mathbf{C}$.

It suffices to show that for vertex $A$ such that $A \rightarrow K$ or $A \leftrightarrow K$ in $\mathcal{H}_j$, $A$ is adjacent to $T$. Then according to the ancestral property of $\mathcal{H}_i$, we directly get the desired result due to $K \rightarrow T$ in $\mathcal{H}_j$.

We discuss the possible cases of the edge between $A$ and $K$ in $\mathbb{M}_i$. If there is $A *\!\!\to K$ in $\mathbb{M}_i$, due to the circle at $K$ on the edge of $K$ and $T$ and the closed property of $\mathbb{M}_i$, $A$ is adjacent to $T$. Hence the result evidently holds.

If there is $A \circ\!\!-\!\!\circ K$ in $\mathbb{M}_i$, we discuss whether $A \in \mathbf{C}$. If not, then $A \in \mathrm{PossDe}(X, \mathbb{M}_i[-\mathbf{C}])$ due to $K \in \mathrm{PossDe}(X, \mathbb{M}_i[-\mathbf{C}])$. Suppose $T$ is not adjacent to $A$, we will prove its impossibility. In this case, we orient $K \to A$ in the second step due to $T \in \mathcal{F}_K \backslash \mathcal{F}_A$, there is thus $K \to A$ in $\mathcal{H}_0$. Considering we do not transform $\to$ to $\leftarrow$ in the whole procedure, there cannot be an edge $A \to K$ in $\mathcal{H}_j$. And since only the directed edge connecting a vertex in $\mathbf{C}$ and a vertex in $\mathrm{PossDe}(X, \mathbb{M}_i[-\mathbf{C}])$ is possibly converted to a bi-directed edge in the process from $\mathcal{H}_0$ to $\mathcal{H}_j$, $A \leftarrow K$ is also not transformed to $A \leftrightarrow K$ due to $A, K \in \mathrm{PossDe}(X, \mathbb{M}_i[-\mathbf{C}])$, so that $A \leftrightarrow K$ cannot be in $\mathcal{H}_j$. Hence when $A \circ\!\!-\!\!\circ K$ in $\mathbb{M}_i$ and $A \notin \mathbf{C}$, there is not an edge $A \to K$ or $A \leftrightarrow K$ in $\mathcal{H}_j$. If $A \in \mathbf{C}$, $A$ is adjacent to $T$ due to $T \in \mathbf{C}$ and Lemma 9. Hence the result holds when $A \in \mathbf{C}$. We conclude that if there is $A \circ\!\!-\!\!\circ K$ in $\mathbb{M}_i$, the result holds.

If there is $A \leftarrow\!\!\circ K$ in $\mathbb{M}_i$, there is $A \leftarrow K$ in $\mathcal{H}_0$. Since we do not add an arrowhead at a vertex in $\mathbf{C}$ in the process of obtaining $\mathcal{H}$ from $\mathcal{H}_0$, and only the directed edge connecting a vertex in $\mathbf{C}$ and a vertex in $\mathrm{PossDe}(X, \mathbb{M}_i[-\mathbf{C}])$ is possibly converted to a bi-directed edge in the process from $\mathcal{H}_0$ to $\mathcal{H}_j$, we only need to consider there is $A \leftrightarrow K$ in $\mathcal{H}_j$, where $A \in \mathbf{C}$. In this case, $A$ is adjacent to $T$ by Lemma 9. The result holds.

For the other cases for the edge between $A$ and $K$ in $\mathbb{M}_i$ except for $A *\!\!\to K$, $A \circ\!\!-\!\!\circ K$, and $A \leftarrow\!\!\circ K$, there cannot be an edge as $A \to K$ or $A \leftrightarrow K$ in $\mathcal{H}_j$. We thus have considered all the possible cases and conclude that if there is an edge $A \to K$ in $\mathcal{H}_j$, there is $A \to T$ in $\mathcal{H}_j$; if there is $A \leftrightarrow K$ in $\mathcal{H}_j$, either $A \to T$ or $A \leftrightarrow T$ is in $\mathcal{H}_j$ according to the balanced property.

(3) In this part, we prove that there is no discriminating path for $K$ on which $T$ is the endpoint adjacent to $K$ in $\mathcal{H}_j$. The proof of this part refers to the proof of (T3) of Theorem 3 by Zhang [35], with modifications due to the additional background knowledge.

Suppose a path $p = (V_0, V_1, \cdots, V_n = K, T)$ which is a discriminating path for $K$. Without loss of generality, suppose $p$ is the shortest path. According to the construction of $\mathrm{Edge}(\mathcal{H}_0)$, there is $K \circ\!\!\to T$ in $\mathbb{M}_{i+1}$. We derive a contradiction by showing that $p$ is already a discriminating path in $\mathbb{M}_i$. Hence there cannot be an edge $K \circ\!\!\to T$ in $\mathbb{M}_i$, otherwise if $i \geq 1$ it will be oriented as $K \to T$ by $\mathcal{R}'_4$ or if $i = 0$ it will be oriented as $K \to T$ or $K \leftrightarrow T$ by $\mathcal{R}_4$ due to the closed property of $\mathbb{M}_i$. There is $V_{n-1} \leftrightarrow K$ in $\mathcal{H}_j$, for otherwise there would be a directed path $K \to V_{n-1} \to T$ from $K$ to $T$ other than the edge $K \to T$ in $\mathcal{H}_j$. It follows that every edge on the subpath from $V_1$ to $K$ is bi-directed in $\mathcal{H}_j$.

Next we will prove that there is an edge $V_0 *\!\!\to V_1$ in $\mathbb{M}_i$. Suppose for contradiction, the edge is either $V_0 \circ\!\!-\!\!\circ V_1$ or $V_0 \leftarrow\!\!\circ V_1$.

**(i).** Suppose $V_0 \circ\!\!-\!\!\circ V_1$ in $\mathbb{M}_i$. There cannot be an edge $V_1 \leftrightarrow V_2$ in $\mathbb{M}_i$, for otherwise there is $V_0 \leftrightarrow V_2$ in $\mathbb{M}_i$ by balanced property of $\mathbb{M}_i$, which contradicts with the shortest discriminating path $p$. Since we do not transform a circle edge in $\mathbb{M}_i$ to a bi-directed edge, the edge between $V_1$ and $V_2$ are either $V_1 \circ\!\!\to V_2$ or $V_1 \leftarrow\!\!\circ V_2$. For the former case, $V_0$ is adjacent to $V_2$, for otherwise $V_0 *\!\!\to V_1 \leftarrow\!\!* V_2$ is identifiable in $\mathcal{P}$ and $\mathbb{M}_i$ since $V_0 *\!\!\to V_1 \leftrightarrow V_2$ in $\mathcal{H}_j$ and $\mathcal{H}_j$ is an MAG Markov equivalent to $\mathcal{H}_0$ which belongs to the MEC represented by $\mathcal{P}$, contradicting with $V_0 \circ\!\!-\!\!\circ V_1$ in $\mathbb{M}_i$. According to the balanced property of $\mathbb{M}_i$, there is $V_0 *\!\!\to V_2$ in $\mathbb{M}_i$ thus there is $V_0 *\!\!\to V_2$ in $\mathcal{H}_j$, in which case there is a shorter discriminating path without $V_1$, contradiction. For the latter case, there is $V_0 \circ\!\!-\!\!\circ V_1 \leftarrow\!\!\circ V_2$ in $\mathbb{M}_i$. As shown by the orientation procedure, we only add an arrowhead at the vertex in $\mathrm{PossDe}(X, \mathbb{M}_i[-\mathbf{C}])$, and we never orient an edge as bi-directed edge in an edge connecting two vertices from $\mathrm{PossDe}(X, \mathbb{M}_i[-\mathbf{C}])$, hence $V_0 *\!\!\to V_1$ and $V_1 \leftrightarrow V_2$ cannot be oriented at the same time in the process of obtaining $\mathcal{H}$ from $\mathcal{H}_0$.

**(ii).** Suppose $V_0 \leftarrow\!\!\circ V_1$. Due to the fact that a bi-directed edge is oriented in $\mathcal{H}_j$ compared to $\mathbb{M}_i$ only if the edge connects a vertex in $\mathrm{PossDe}(X, \mathbb{M}_i[-\mathbf{C}])$ and a vertex in $\mathbf{C}$, and the fact that an arrowhead is added only at the vertex in $\mathrm{PossDe}(X, \mathbb{M}_i[-\mathbf{C}])$, there is $V_0 \in \mathbf{C}$ and $V_1 \in \mathrm{PossDe}(X, \mathbb{M}_i[-\mathbf{C}])$. And due to $T \in \mathbf{C}$ and the non-adjacency of $T$ and $V_0$, there is a contradiction with the condition that $\mathbb{M}_i[\mathbf{C}]$ is complete in Lemma 9.

We conclude there is $V_0 *\!\!\to V_1$ in $\mathbb{M}_i$. The remaining part is to prove by induction that for every $1 \leq i \leq n-1$, $V_i$ is a collider and a parent of $T$ in $\mathbb{M}_i$. $V_1 \to T$ is evident due to the non-

adjacency of $V_0$ and $T$. Note $T \in \mathbf{C}$ and $V_1 \to T$ in $\mathbb{M}_i$, thus $V_1 \notin \mathrm{PossDe}(X, \mathbb{M}_i[-\mathbf{C}])$ due to $\mathrm{PossDe}(X, \mathbb{M}_i[-\mathbf{C}]) \cap \mathrm{Pa}(\mathbf{C}, \mathbb{M}_i) = \emptyset$ as Lemma 10. There cannot be an edge $V_1 \to V_2$ in $\mathbb{M}_i$ because the edge cannot be oriented as $V_1 \leftrightarrow V_2$ in $\mathcal{H}_j$. If there is not a collider at $V_1$ in $\mathbb{M}_i$, there is $V_1 \circ\!\!\to V_2$. We orient it to bi-directed edges only if $V_1 \in \mathrm{PossDe}(X, \mathbb{M}_i[-\mathbf{C}])$, contradiction. Hence the collider is identifiable in $\mathbb{M}_i$. Similarly, we could prove $V_2 \to T$ in $\mathbb{M}_i$. Due to $T \in \mathbf{C}$ and $\mathrm{PossDe}(X, \mathbb{M}_i[-\mathbf{C}]) \cap \mathrm{Pa}(\mathbf{C}, \mathbb{M}_i) = \emptyset$, $V_2 \notin \mathrm{PossDe}(X, \mathbb{M}_i[-\mathbf{C}])$, thus $V_1 \leftrightarrow V_2 \leftarrow\!\!*V_3$ is identifiable in $\mathbb{M}_i$ since arrowhead is added at only the vertex in $\mathrm{PossDe}(X, \mathbb{M}_i[-\mathbf{C}])$. By such way, we prove that the path is a discriminating path for $K$ in $\mathbb{M}_i$. Thus there cannot be an edge $K \circ\!\!\to T$ in $\mathbb{M}_i$, otherwise it will be oriented as $K \to T$ by $\mathcal{R}_4'$ if $i \geq 1$ and oriented as $K \to T$ or $K \leftrightarrow T$ if $i = 0$ since $\mathbb{M}_i$ is closed under the orientation rules, contradicting with the fact that there is $K \circ\!\!\to T$ in $\mathbb{M}_{i+1}$.

Hence, we conclude that $\mathcal{H}$ is an MAG Markov equivalent to $\mathcal{H}_0$. It is evident that $\mathcal{H}$ has the non-circle marks in $\mathbb{M}_{i+1}$. Since $\mathcal{H}_0$ belongs to the MEC represented by $\mathcal{P}$, $\mathcal{H}$ also belongs to the MEC. We conclude that $\mathcal{H}$ is an MAG consistent to $\mathcal{P}$ and the local BK regarding $V_1, \cdots, V_{i+1}$. The proof in this part completes.

Hence, according to the result (**A**), $\mathcal{H}_0$ can be seen as an MAG obtained from $\mathbb{M}_i$ by transforming $\circ\!\!\to$ to $\to$ and the circle component into a DAG without new unshielded colliders in $\mathbb{M}_i$. With the inverse negative proposition of the result (**B**), if $\mathcal{H}$ is not an MAG consistent to $\mathcal{P}$ and the local BK regarding $V_1, \cdots, V_{i+1}$, then $\mathcal{H}_0$ is not an MAG consistent to $\mathcal{P}$ and the local BK regarding $V_1, \cdots, V_i$, which can be obtained from $\mathbb{M}_{i-1}$ by transforming $\circ\!\!\to$ to $\to$ and the circle component into a DAG without new unshielded colliders in $\mathbb{M}_{i-1}$. Repeat the process above, we can conclude that there is a graph obtained from $\mathcal{P}$ by transforming $\circ\!\!\to$ to $\to$ and the circle component into a DAG without new unshielded colliders that is not MAG consistent to $\mathcal{P}$, which contradicts with Theorem 2 of Zhang [35]. We get the desired result. $\qquad\square$

**Lemma 15.2.** *Suppose there is an edge $A \circ\!\!\to B$ in the PMG $\mathbb{M}_{i+1}$ in Thm. 1, then there is an MAG $\mathcal{M}_1$ consistent to $\mathcal{P}$ and local BK regarding $V_1, \cdots, V_{i+1}$ with $A \leftrightarrow B$.*

*Proof.* This part totally follows Theorem 3 of Zhang [35] with the results we have proved before. Hence we only show the sketch. We take $\mathbb{M}_{i+1}$ as the $\mathcal{P}_{AFCI}$ of Zhang [35]. Note we do not consider selection bias in this paper. Hence the cases of $\mathbf{P}_2, \mathbf{P}_3, \mathbf{P}_4$ (Lemma A.2, Lemma A.4, Lemma A.5) of Zhang [35] will not happen. And $\mathbf{P}_1$, *i.e.*, the balanced property, has been proved to hold in $\mathbb{M}_{i+1}$ according to Lemma 14. With the balanced property, Lemma B.1-Lemma B.18 of Zhang [35], which are sufficient to prove Theorem 3 of Zhang [35], also hold in $\mathbb{M}_{i+1}$ because there are not other conditions involved. As proved by Lemma 15.1, we prove that when we transform the $\circ\!\!\to$ edges to $\to$, and orient the circle component into a DAG without new unshielded colliders based on $\mathbb{M}_{i+1}$, we can always obtain an MAG consistent to $\mathcal{P}$ and local BK regarding $V_1, \cdots, V_{i+1}$. It plays the roles of Theorem 2 of Zhang [35]. We can construct a graph $\mathcal{H}$ with $A \leftrightarrow B$ by the same procedure of Theorem 3 of Zhang [35] and prove $\mathcal{H}$ is an MAG that is Markov equivalent to an MAG $\mathcal{H}_0$ obtained from $\mathbb{M}_{i+1}$ by transforming $\circ\!\!\to$ edges to $\to$ and transforming the circle component in $\mathbb{M}_{i+1}$ into a DAG $\mathcal{D}_{A \circ\!\!\to B}$ defined in Theorem 3. According to Lemma 15.1, $\mathcal{H}_0$ is an MAG in the MEC represented by $\mathcal{P}$. Hence $\mathcal{H}$ is an MAG in the MEC represented by $\mathcal{P}$. And since $\mathcal{H}$ has the non-circle edges in $\mathbb{M}_{i+1}$, $\mathcal{H}$ is an MAG with $A \leftrightarrow B$ consistent to $\mathcal{P}$ and local BK regarding $V_1, \cdots, V_{i+1}$. $\qquad\square$

**Theorem 1.** *Given $i$, suppose $\mathbb{M}_s, \forall s \in \{0, 1, \ldots, i\}$ satisfies the five following properties:*

*(**Closed**) $\mathbb{M}_s$ is closed under the orientation rules.*

*(**Invariant**) The arrowheads and tails in $\mathbb{M}_s$ are invariant in all the MAGs consistent to $\mathcal{P}$ and BK regarding $V_1, \ldots, V_s$.*

*(**Chordal**) The circle component in $\mathbb{M}_s$ is chordal.*

*(**Balanced**) For any three vertices $A, B, C$ in $\mathbb{M}_s$, if $A *\!\!\to B \circ\!\!-\!* C$, then there is an edge between $A$ and $C$ with an arrowhead at $C$, namely, $A *\!\!\to C$. Furthermore, if the edge between $A$ and $B$ is $A \to B$, then the edge between $A$ and $C$ is either $A \to C$ or $A \circ\!\!\to C$ (i.e., it is not $A \leftrightarrow C$).*

*(**Complete**) For each circle at vertex $A$ on any edge $A \circ\!\!-\!* B$ in $\mathbb{M}_s$, there exist MAGs $\mathcal{M}_1$ and $\mathcal{M}_2$ consistent to $\mathcal{P}$ and BK regarding $V_1, \ldots, V_s$ with $A \leftarrow\!\!* B \in \mathbf{E}(\mathcal{M}_1)$ and $A \to B \in \mathbf{E}(\mathcal{M}_2)$.*

*Then the PMG* $\mathbb{M}_{i+1}$ *obtained from* $\mathbb{M}_i$ *with* $BK(V_{i+1})$ *by Alg. 1 also satisfies the five properties.*

*Proof.* The closed, invariant, chordal, balanced, complete properties of $\mathbb{M}_{i+1}$ are proved by Lemma 5, 6, 13, 14, 15. □