# OpenReview forum: "Sound and Complete Causal Identification with Latent Variables Given Local Background Knowledge"
_NeurIPS.cc/2022/Conference — NeurIPS 2022 Accept_

### Official Review · Reviewer_Zw4J · 2022-07-08

**Rating:** 6
**Confidence:** 3
**Soundness:** 3 good
**Presentation:** 2 fair
**Contribution:** 3 good

**Summary:**

For a given PAG, the paper presents a sound and complete procedure to incorporate local background knowledge (BK). For a given node X, the local BK the paper considers is knowledge of the edge marks at X, i.e., the circle marks at X in the given PAG are converted to an arrowhead or a tail using the BK. Then the procedure orients further edges/nodes in the PAG. For a given set of nodes with BK, the procedure sequentially incorporates the BK for each node one at a time.
Next, the paper presents an algorithm to decide which nodes to intervene on (interventions are essentially equivalent the BK) with the goal of learning the MAG with a small number of interventions.



**Questions:**

Comparison to learning PAGs with interventional data:
There is some work on incorporating interventional data into PAGs. As an example, see [1], which provides a characterization as well as a learning algorithm for causal discovery under unknown interventions (with known interventions being a special case). This algorithm is also sound and complete. BK is not necessarily the same as soft interventions (which can reveal more information). But to me, they seem very closely related as interventions are one way of obtaining BK, so it seems to me that algorithms that are sound and complete in incorporating interventions can also be leveraged for BK. Can the authors comment on how their work is related to this line of work?

Line 156:
What do you mean by "algorithm can be achieved"? What does it mean for an algorithm to be achieved?


[1]  Jaber, A., Kocaoglu, M., Shanmugam, K., & Bareinboim, E. (2020). Causal discovery from soft interventions with unknown targets: Characterization and learning. Advances in neural information processing systems, 33, 9551-9561.

**Limitations:**

I have addressed this in the previous sections.

**Strengths And Weaknesses:**

Originality, quality, and significance:
The primary contribution of the work is to propose two new orientation rules that allow for sound and complete incorporation of BK for a given node V_i (in Alg. 1). I think the work is significant as it proposes a way to further narrow down the MEC beyond what can be learned using only observational data. And incorporating this specific kind of BK is also novel (although I think more comparison is needed to related literature --- please see my remarks in the "Questions" section).

Clarity:
Overall, I think the paper is hard to read. I would have appreciated some more examples to illustrate how the algorithm work. It would have been nice if the proof sketches were explained with concrete examples of why the two new rules are sound and complete (especially for Thm. 2).

---

> ### Author Response · Authors · 2022-08-02
> **Response to Reviewer Zw4J**
>
> Thank you for your valuable comments! We will revise our paper accordingly. And we will also give more examples for the illustrations in Line 178-187 about the intuition of the steps of Alg. 1, and provide some concrete examples of the reason that the proposed orientation rules (Zhang’s rules with additional $R_4’,R_{11}$) are complete.
>
> Q1. About Line 156 the meaning of “algorithm can be achieved”: We wanted to express that in the second part of the proof, we prove that the proposed orientation rules can orient the same edges as Alg. 1. The ultimate objective of the whole proof is to prove that the orientation rules are sound and complete to orient a PAG with local BK, which is hard to prove directly. Hence, we introduce Alg. 1 as a bridge, and divide the whole proof process into two steps. In the first step, we prove that Alg. 1 is sound and complete to orient the PAG with local BK. In the second step, we prove that Alg. 1 orients the same edges as the proposed orientation rules. These two steps together proved the soundness and completeness of the orientation rules. We will make it clearer in the revised version.
>
> Q2. About connection to Jaber et al.: We will illustrate it in revised versions.
> From the viewpoint of *problem*: (i) The algorithms (such as Jaber et al.) that are sound and complete in incorporating interventions are not necessarily able to be leveraged for local BK. **The input of the two methods are different**. In addition to the PAG, the input of our method is the local marks regarding $X$ (only $X$ is considered for example), the input in their work is the distribution under intervention on $X$. The inter. distribution can imply the local marks, but not vice versa. Hence, local BK is a weaker requirement than inter. distributions. That implies, although the compared method is complete with inter. distribution, it is not necessarily applicable with local BK. For example, according to human experience regarding $X$, we know the marks at $X$, but human experience cannot give us the inter. distribution. In this case, Line 18 of Alg. 1 of Jaber et al. cannot be tested (this step is irrelevant to whether inter. target is known and whether we learn $\mathcal{P}$ in advance). Hence for this kind of BK, Alg. 1 of Jaber et al. cannot be executed. In summary, the input of the two methods are not identical. (ii) Then even if we *only consider the local BK from inter. data*, if the inter. distribution is fully available such that Line 18 of Alg. 1 of Jaber et al. can be correctly tested, their method can obtain the desired graph. But we want to emphasize that an advantage of considering local BK is that this kind of input information does not need large amount of inter. data to estimate the inter. distribution (As pointed out by Reviewer Wrk3, the inter. data is often expensive.). The reason is that our method based on local BK only needs to learn the local marks at $X$ by some two-sample tests such as whether $P(Z|do(X))=P(Z)$ ($X$ and $Z$ are singleton variables). It is not hard to achieve. The samples required in the two-sample test to ensure low test error is far fewer than that in testing Line 18 of Alg. 1 of Jaber et al. because of the possibly high-dimensional $\mathbf{W}$ and the intrinsic hardness of conditional independence tests.
>
> From the viewpoint of *techniques*: To prove completeness, both methods construct MAGs with each circle being an arrowhead and a tail. Jaber et al. constructs an MAG over $\mathbf{V}\cup \mathbf{F}$ with obser. and inter. distributions, while we construct an MAG over $\mathbf{V}$ with obser. distribution and local marks regarding some variables. Even if we put aside the difference of input, the construction of Jaber et al. is essentially constructing an MAG over $\mathbf{V}\cup \mathbf{F}$ based on a special kind of local BK regarding $\mathbf{F}$ where all the edges regarding $\mathbf{F}$ are out of $\mathbf{F}$ (i.e., the local BK dictates that the marks at $\mathbf{F}$ are **tails**). There are also some similar constructions such as Lemma 7.6 of Maathuis et al. and Lemma 43 of Perković et al.. The construction in our method is constructing an MAG over $\mathbf{V}$ based on $\mathcal{P}$ and local BK regarding some vertices where there could be **additional arrowheads** dictated by the local BK. The additional arrowheads at given variables take the great challenge to the construction. It is the reason that our method introduces two complex steps - the first and second steps of Alg. 1, in the construction, in contrast to previous construction methods [Maathuis et al., Perković et al., Jaber et al.] developed directly based on Thm. 2 of [31].
>
> - A generalized back-door criterion. Marloes H. Maathuis and Diego Colombo.
> - Complete Graphical Characterization and Construction of Adjustment Sets in Markov Equivalence Classes of Ancestral Graphs. Emilija Perković, Johannes Textor, Markus Kalisch and Marloes H. Maathuis.

---

> > ### Comment · Reviewer_Zw4J · 2022-08-07
> > **Thanks for the response**
> >
> > Thanks for the response regarding the comparison to Jaber et al. I will keep my current score.

---

### Official Review · Reviewer_Wrk3 · 2022-07-10

**Rating:** 7
**Confidence:** 3
**Soundness:** 3 good
**Presentation:** 3 good
**Contribution:** 3 good

**Summary:**

Nonparametric causal discovery can identify the true DAG up to some undertainty. The uncertainty can be eliminated only by incorporating background knowledge (BK). Under causal sufficiency assumption, Meek rules established a complete rule to orient the CPDAG to DAG with BK. While in the presence of latent variables, authors provide the first sound and complete rules to orient the PAG. The orientation rules are based on local form of BK. With these rules, authos propose an active learning framework, aiming to orient a given PAG with the minimum number of interventions (BKs). Since it is hard to count the number of MAGs in some consistent space, authors further equipped the method with the adaptive rejection Metropolis-Hastings sampling. Experimental results demonstrate the effectiveness of the proposed framework.

**Questions:**

1. **local BK**: now BK is defined as _all marks_ on a variable X - which seems expensive - and usually BK is given as some specific direct edges. Could the authors please give more motivation on why this BK is defined over some node, instead of edge? If some BK is given in the edge form, could it also be incorporated into this framework?
2. **Latent**: by _with latent variables_, is it assumed that all latent variables are exogenous (or mutually independent), similar as assumptions in FCI, so that the Markov equivalence class can be represented as a PAG. And thus, in experiments, L348 "take three nodes as latent variables and the others as observed ones" - are they taken randomly or with some criteria?
3. **Correctness of BK**: L134: "The correctness indicates that there exists an MAG consistent to P and the BK." So the authors also assume that the learned P at the first stage is correct. What if P is incorrect (which is often the case), and the given BK has confict with P - is there any correction rule to prevent cascading error?
4. **About originality**: the active learning framework (maximal entropy criterion, rejection metropolis sampling etc, section 4) seems to be a classic paradigm in combinational search. Is this originally proposed by authors, or very similar to some existed methods on orientation rules?
5. **Accuracy difference in Table 1**: why there is accuracy difference between MCMC and random? Since the orientation rules are deterministic, given the same PAG, the maximum oriented graph should be exactly the same, with only the difference in the number of interventions required. Where does the accuracy difference come from? What if the starting PAG is exactly the true PAG (asymptotic result, not by FCI)?
6. L86: letters (V, E) should be bold face.

**Limitations:**

The problem setup is quite clear, and the authors have adequately addressed the limitations. I only have some questions above, regarding assumptions on latent, and assumptions on correct P and BK.

**Strengths And Weaknesses:**

+ Pros:
  - **Clarity.** The paper is generally well-written and organized. The problem definition is set clear, the preliminaries are described in a detailed and clean way, and theoretical analyses are also well carried out.
  - **Technical quality of the proposed rules.** The proposed orientation rules are technically dense. I haven't read the rules (section 3) in detail, and could not ensure the correctness. But I tried on some simple graphs example, and the rules appear to be correctly identifiable to me.

+ Cons: Overall this is good to me. Currently I could not give any specific weaknesses of this paper, but only some questions. See the questions section.

---

> ### Author Response · Authors · 2022-08-02
> **Response to Reviewer Wrk3**
>
> Thank you for your insightful questions and positive evaluation! We will revise the paper accordingly.
>
> Q1. About local BK: Defining the local BK over nodes is motivated by the fact that BK can be obtained through interventions, which are often performed on certain nodes instead of edges. When we intervene on some variable $X$, we can learn the marks at $X$ on the edge with $Z$ by testing whether $P(Z|do(X))$ equals $P(Z)$.
>
> For general BKs, including BK in edge forms, the rules in this paper are sound, hence can also be exploited. But currently we are not sure if the rules we proposed are complete for general BK, although we have not found any incomplete counterexample.
>
> Q2. About latent: The latent variables in the experiments are taken randomly. To the best of our knowledge, FCI does not need the assumption that all latent variables are mutually independent. Consider a DAG $A\rightarrow B\rightarrow C\rightarrow D$. If $B$ and $C$, which are not mutually independent, are latent variables, then by FCI we learn $A\circ\hspace{-1.4mm}-\hspace{-1.4mm}\circ C$. It is still a valid PAG. And the MAG representing the DAG with observable variables is $A\rightarrow D$.
>
> Q3. About correctness of MEC: In this paper, to focus on the orientation with incorporating BK, we assume the correctness of Markov equivalence class as related methods [28,32]. If we further consider the cases that some mistakes may happen in $\mathcal{P}$ *but assume local BK is always correct*, we can exploit a theoretical result in one of our recent work under review, which given the necessary and sufficient conditions for the existence of MAGs consistent to $\mathcal{P}$ and local BK. For example, with the local BK, if the necessary and sufficient conditions are not fulfilled, it implies that there must be some mistakes in $\mathcal{P}$ because there should exist an MAG consistent to the correct $\mathcal{P}$ and local BK. In this case, we need to collect more observational data, or by other ways, to re-learn a PAG. In general, we feel that it is hard to present the correction rules for preventing the cascading error, because when there are conflicts, there are many kinds of marks corrections even skeleton corrections which can lead to eliminating the conflicts. From the theoretical viewpoint, we cannot determine where the mistakes happen (which correction is consistent to the true PAG) without further assumptions. In practice, we can possibly exploit some heuristic ideas such as taking the correction which changes the least edges in $\mathcal{P}$ to eliminate the conflicts as the correction that is consistent to the true PAG.
>
>
> Q4. About Originality: The key contribution is the sound and complete orientation rules in the presence of latent confounders, which make it possible to conduct a complete active learning framework. Entropy criterion was applied in causal sufficiency setting [20]. Our work is the first to extend the criterion to causal insufficiency setting, and in light of the large space of MAGs compared to DAGs, we introduce rejection sampling for MAGs.
>
> Q5. About accuracy difference: The difference is due to the type I and type II errors of two-sample tests. When we intervene on $X$, for example, we learn the marks at $X$ on an edge between $X$ and $Z$ by testing whether $P(Z|do(X))$ is equal to $P(Z)$. Because the intervention variables can be different for different methods (MCMC and random), different strategies suffer different test errors in real implementation, which causes the difference.

---

> > ### Comment · Reviewer_Wrk3 · 2022-08-09
> > **RE Response to Reviewer Wrk3**
> >
> > Thank you for answering my questions in detail. I like this paper. I will maintain my recommendation of acceptance.

---

### Official Review · Reviewer_BMb7 · 2022-07-11

**Rating:** 7
**Confidence:** 3
**Soundness:** 3 good
**Presentation:** 3 good
**Contribution:** 4 excellent

**Summary:**

This paper presents sound and complete orientation rules to incorporate local causal background knowledge to PAGs and applies these rules to active learning. Here, the local causal background knowledge regarding a variable $X$ refers to the causal relations about X with any adjacent variable. Note that, this paper assumes no selection bias.

**Questions:**

I also only have some questions for the authors
1. The authors assume the local BK is correct, meaning that the orientation rules can only be applied when the local BK is correct. When the local BK is not correct, will applying the orientation rules cause some mistakes (such as cycles or new colliders), or we can still get a PAG?
2. Regarding the MH sampling, it seems that Proposition 2 only gives a method to construct an equivalent MAG from a given one. Can any two Markov equivalent DAGs be transformed into each other in a limited number of these transformations? Is it possible that a sample MAG from $S({\cal M}_{t-1})$ is not consistent with the given BK (line 3, Algorithm 2)? What is the computational complexity of Algorithm 2?


**Limitations:**

Not applicable.

**Strengths And Weaknesses:**

This paper is clearly written and the contributions are novel. The authors prove sound and complete orientation rules to orient a PAG with local background knowledge. Although I did not check the proof, the sketch seems reasonable. I only have one suggestion: it would be better to explain the steps to construct the PAGs in the example presented in Figure 1 in more detail.

---

> ### Author Response · Authors · 2022-08-02
> **Response to Reviewer BMb7**
>
> Thank you for your careful reading and valuable questions! We will explain the steps of Alg. 1 with the examples in Fig. 1.
>
> Q1. About the case that local BK is not correct: In general, wrong BK can introduce some mistakes. For example, consider a PAG $A\circ\hspace{-1.4mm}-\hspace{-1.4mm}\circ B\circ\hspace{-1.4mm}-\hspace{-1.4mm}\circ C$, if the local BK regarding $B$ tells us that there are both arrowheads at $B$ on the edge between $B$ and $A$ and the edge between $B$ and $C$, the obtained graph is evidently invalid since there are additional v-structures introduced. Although wrong BK may be ubiquitous in real-world problems, to make a step towards understanding the impact of BK in causality, assuming BK is correct is still a common practice in related works [28,32].  However, we want to point out that in some cases, it is possible that we identify the existence of mistakes of local BK if we only assume the correctness of PAG, which can help analysts realize that they should re-evaluate the correctness of local BK before performing causal analysis. In a paper under review in one of our recent work, we gave necessary and sufficient conditions for the existence of MAGs consistent to $\mathcal{P}$ and local BK (suppose it is $BK(X)$). If the local marks at $X$ dictated by the local BK do not fulfill the conditions, there cannot exist MAGs consistent to $\mathcal{P}$ and the local BK, which implies that the local BK is wrong.
>
>
> Q2. About MH Algorithm: It is a very penetrating question! Two Markov equivalent MAGs can be transformed to each other in limited number transformation, which is shown by Thm. 3 of [37]. And due to Prop. 2 and the fact proved by Lemma 15.1 that the MAG obtained on Line 1 of Alg. 2 is an MAG consistent to the given BK and PAG, all sampled MAGs are consistent to the PAG. However, the sampled MAGs are not necessarily consistent to the given BK. Hence, in real implementation of our method, after we sample $L$ MAGs, we have an additional step to judge whether each sampled MAG is consistent to the local BK (has the non-circle marks dictated by the local BK), and we only keep the $L’$ MAGs that are consistent and calculate the entropy with these $L’$ MAGs. We will make it clear in Alg. 2 in the revised version to avoid misunderstanding.
>
> About the complexity of Alg. 2: the running time of applying Prop. 2 mainly depends on the time spent on determining if the third condition is met, which is $O(\lvert V\rvert\lvert E\rvert)$ (See footnote 20 of [31]). There are at most $\lvert E\rvert $ edges that can be transformed. Hence the complexity of Line 3 of Alg. 2 is $O(\lvert V\rvert\lvert E\rvert^2)$. Since the complexity of Alg. 2 mainly depends on Line 2 to Line 6, the total complexity of Alg. 2 is $O(L\lvert V\rvert\lvert E\rvert^2)$.

---

> > ### Comment · Reviewer_BMb7 · 2022-08-08
> > **Reply to Authors**
> >
> > Thank you for your clarification. All my concerns have been addressed.

---

### Meta-Review · Area_Chair_RnU7 · 2022-08-26

**Recommendation:** Accept
**Confidence:** Certain

**Metareview:**

This paper presents sound and complete orientation rules to incorporate local causal background knowledge along with algorithms implementing these rules. Reviewers were universally appreciative of the contributions and in favor of acceptance.

**Award:**

No

---

### Decision · Program_Chairs · 2022-09-14

Accept